

# Evaluation of air-soil temperature relationships simulated by land surface models during winter across the permafrost region

**Wenli Wang[1+], Annette Rinke[1,2+], John C. Moore[1], Duoying Ji[1*], Xuefeng Cui[3], Shushi Peng[4,17,18], David M. Lawrence[5], A. David McGuire[6], Eleanor J. Burke[7], Xiaodong Chen[8], Christine Delire[9], Charles Koven[10], Andrew MacDougall[11], Kazuyuki Saito[12,15], Wenxin Zhang[13, 19], Ramdane Alkama[9,16], Theodore J. Bohn[8], Philippe Ciais[18], Bertrand Decharme[9], Isabelle Gouttevin[4], Tomohiro Hajima[12], Gerhard Krinner[4,17], Dennis P. Lettenmaier[8], Paul A. Miller[13], Benjamin Smith[13], Tetsuo Sueyoshi[14]**

[1]College of Global Change and Earth System Science, Beijing Normal University, Beijing, China

[2]Alfred Wegener Institute Helmholtz Centre for Polar and Marine Research (AWI), Potsdam, Germany

[3]School of System Science, Beijing Normal University, Beijing 100875, China

[4]CNRS, LGGE, 38041 Grenoble, France

[5]National Center for Atmospheric Research, Boulder, USA

[6]U.S. Geological Survey, Alaska Cooperative Fish and Wildlife Research Unit, University of Alaska Fairbanks, Fairbanks, AK, USA

[7]Met Office Hadley Centre, Exeter, UK

[8]School of Earth and Space Exploration, Arizona State University, Tempe, AZ, USA

[9]GAME, Unit émixte de recherche CNRS/Meteo-France, Toulouse cedex, France

[10]Lawrence Berkeley National Laboratory, Berkeley, CA, USA

[11]School of Earth and Ocean Sciences, University of Victoria, Victoria, BC, Canada



[12]Japan Agency for Marine-Earth Science and Technology, Yokohama, Japan
[13]Department of Physical Geography and Ecosystem Science, Lund University, Lund, Sweden
[14]National Institute of Polar Research, Tachikawa, Japan
[15]University of Alaska Fairbanks, Fairbanks, AK, USA
[16]l'Institute for Environment and Sustainability (IES), Ispra, Italy
[17]Université Grenoble Alpes, LGGE, 38041 Grenoble, France
[18]LSCE, CEA/CNRS/UVSQ, Saclay, France
[19]Center for Permafrost (CENPERM), Department of Geosciences and Natural Resource
Management, University of Copenhagen, DK-1350 Copenhagen, Denmark
*+ these authors contributed equally to this work*
*\* correspondence to*: Duoying Ji (duoyingji@bnu.edu.cn)



**Abstract.** A realistic simulation of snow cover and its thermal properties are important for accurate modelling of permafrost. We analyze simulated relationships between air and near-surface (20 cm) soil temperatures in the Northern Hemisphere permafrost region during winter, with a particular focus on snow insulation effects in nine land surface models and compare them with observations from 268 Russian stations. There are large across-model differences as expressed by simulated differences between near-surface soil and air temperatures, ($\Delta T$), of 3 to 14 K, in the gradients between soil and air temperatures (0.13 to 0.96 ℃/℃), and in the relationship between $\Delta T$ and snow depth. The observed relationship between $\Delta T$ and snow depth can be used as a metric to evaluate the effects of each model's representation of snow insulation, and hence guide improvements to the model's conceptual structure and process parameterizations. Models with better performance apply multi-layer snow schemes and consider complex snow processes. Some models show poor performance in representing snow insulation due to underestimation of snow depth and/or overestimation of snow conductivity. Generally, models identified as most acceptable with respect to snow insulation simulate reasonable areas of near-surface permafrost (12-16 million km$^2$). However, there is not a simple relationship between the quality of the snow insulation in the acceptable models and the simulated area of Northern Hemisphere near-surface permafrost, likely because several other factors such as differences in the treatment of soil organic matter, soil hydrology, surface energy calculations, and vegetation also provide important controls on simulated permafrost distribution.



## 1 Introduction

Present-day permafrost simulations by global climate models are limited and future projections contain high, model-induced uncertainty (e.g., Slater and Lawrence, 2013; Koven et al., 2013). Most of the model biases and across-model differences in simulating permafrost area are due to biased simulations of the atmosphere (air temperature, precipitation), land (snow, soil temperature) and their coupling. In winter, the snow insulation effect is a key process for the air-soil temperature coupling. Its strength depends on both the snow quantity (depth, areal coverage) and snow quality (density, conductivity) (see overview by Zhang, 2005). Many individual model studies have shown the strong impact of a few snow parameters on soil temperature simulations (e.g., recently, Dutra et al., 2012; Gouttevin et al., 2012; Essery et al., 2013; Wang et al., 2013; Jafarov et al., 2014). Most importantly, consideration of wet snow metamorphism and snow compaction, improved snow thermal conductivity and multi-layer snow schemes improve the simulation of snow dynamics and soil temperature. Parameterizations that take into account snow compaction (e.g. related to overburden pressure, thermal metamorphism and liquid water) work better than simpler schemes such as an exponential increase of density with time (Dutra et al., 2010). The influence of snow thermal conductivity on soil regime has been demonstrated by many model studies (e.g., Bartlett et al., 2006; Saha et al., 2006; Vavrus, 2007; Nicolsky et al., 2007; Dankers et al., 2011; Gouttevin et al., 2012). Winter soil temperature can change by up to 20 K simply by varying the snow thermal conductivity by 0.1-0.5 W m$^{-1}$ K$^{-1}$ (Cook et al., 2008). The snow insulation effect also plays an important role for the Arctic soil temperature response to climate change (e.g., reduced soil warming due to shallower snow versus enhanced soil warming due to shorter snow season) (e.g., Lawrence and Slater, 2010), and therefore for future near-surface permafrost thawing and soil carbon vulnerability (e.g., Schuur et al., 2008). The model skill in atmosphere-soil coupling with the concomitant snow cover in the Arctic is an important factor in the assessment of limitations and uncertainty of carbon mobility estimates (Schaefer et al., 2011).

The Snow Model Intercomparison Project (Snow MIP) (Essery et al., 2009) and the Project for Intercomparison of Land-Surface Parameterization Schemes (PILPS) Phase 2e (Slater et al., 2001) examined the snow simulations of an ensemble of land-surface schemes for the mid-latitudes. However, until now there has been no attempt to evaluate the air-soil temperature relationship in the Northern Hemisphere permafrost region and the detailed role



of snow depth therein across an ensemble of models. In such an investigation, a first suitable approach is the evaluation of stand-alone (off-line) land surface models (LSMs). The retrospective (1960-2009) simulations from the model integration group of the Permafrost Carbon Network ("PCN"; http://www.permafrostcarbon.org) provide an opportunity to evaluate an ensemble of nine state-of-the-art LSMs. Here, the LSMs are run with observation-based atmospheric forcing, meaning that snow depth is not influenced by biases in the atmospheric forcing in a coupled model set-up. The evaluation of the offline modeled air temperature - snow depth - near-surface soil temperature relationship in winter is therefore important for revealing a model's skill in representing the effects of snow insulation.

Most of the LSMs participating in PCN are the land-surface modules of Earth System Models (ESMs) participating in the Coupled Model Intercomparison Project (CMIP5; http://cmip-pcmdi.llnl.gov/cmip5/) although in some cases different versions were used for PCN and CMIP5 simulations. Thus, the results we present can guide the corresponding evaluation of these ESMs, though analysis of coupled model results requires consideration of couplings between model components and is necessarily more complex.

The scope of the present study is to analyze the extent to which the ensemble of PCN models can reproduce the observed relationship between air and near-surface soil temperatures in the Northern Hemisphere permafrost region during winter, with a particular focus on the snow insulation effect. For the latter we analyze the impact of snow depth on the difference between near-surface soil and air temperatures. Our related key questions are: How well do the models represent the observed magnitude and spatial pattern of the snow depth control over the air-soil temperature difference in winter? What is the range of these simulated relationships across the model ensemble? To the extent possible, we try to relate the performance level to the model's snow schemes. With this aim in mind, a simultaneous analysis of simulated air and near-surface soil temperatures, and snow depth is presented and compared with those from a novel set of Russian station observations. We focus here on a comprehensive Russian station data set because this has been compiled within PCN, and it is hard to find other station data sets which provide simultaneous observations of both air and soil temperatures as well as snow depth over a long period.

In Sect. 2, we describe the model simulations, the station observations used for evaluation, and the analysis methods. In Sect. 3, we present a detailed analysis of near-surface air




temperature - snow depth - soil temperature relationships in winter. In Sect. 4, we discuss the roles of atmospheric forcing and model processes. We summarize our findings and present conclusions in Sect. 5.

## 2  Data and Analysis

### 2.1  Models

We use data from nine LSMs participating in the PCN, including CLM4.5, CoLM, ISBA, JULES, LPJ-GUESS, MIROC-ESM, ORCHIDEE, UVic, and UW-VIC. For detailed information about the models and simulations we refer to Rawlins et al. (2015) and Peng et al. (2015). The total soil depth for soil thermal calculations ranges from 3 m (divided in 8 layers) in LPJ-GUESS to 250 m (divided in 14 layers) in UVic. The soil physical properties differ among the models as well, and four of them (CLM4.5, ISBA, UVic, UW-VIC) include organic horizons. Three models (ISBA, LPJ-GUESS, UW-VIC) did not archive soil sub-grid results and provide area-weighted ground temperature (i.e. averaged over wetlands and vegetated areas, and in some cases lake fractions).

Table 1 lists relevant snow model details. One model (UVic) uses an implicit snow scheme which replaces the upper soil column with snow-like properties, i.e. the near-surface soil layer takes the temperature of the air-snow interface. The other models use separate snow layers on top of the ground, either a single bucket (LPJ-GUESS, UW-VIC) or multi-layer snow schemes (CLM4.5, CoLM, ISBA, JULES, MIROC-ESM, ORCHIDEE). Snow insulation is explicitly considered in all models; increasing snow depth increases the insulation effect. Many models consider the effect of varying snow density on insulation (Table 1). This is parameterized by a snow conductivity-density relationship that describes how, as snow density increases, thermal conductivity increases, thereby reducing the snow insulation. Some of the models (LPJ-GUESS, MIROC-ESM, ORCHIDEE, UVic) use a fixed snow density, consider only dry snow and no compaction effects, while others represent liquid water in snow and different processes for snow densification such as mechanical compaction, and thermal and destructive metamorphism (Table 1).

The simulations were generally run for the period 1960-2009, although some simulations were stopped a few years earlier. Each model team was free to choose appropriate driving data sets for weather and climate, atmospheric $CO_2$, nitrogen deposition, disturbance, land cover, soil texture, etc. However, the climate forcing data (surface pressure, surface incident



longwave and shortwave radiation, near-surface air temperature, wind and specific humidity,
rain and snowfall rates) are from gridded observational datasets (e.g. CRUNCEP, WATCH)
(SI Table 1). The exception is MIROC-ESM, which was run as a fully-coupled model, forced
by its own simulated climate. Mean annual temperature of the MIROC-ESM simulations for
the permafrost region were within the range (-7.2 to 2.2 °C) of the other forcing data sets used
in this study and the trend in near-surface air temperature (+0.03 °C yr$^{-1}$) was the same for all
forcing data sets. However, MIROC-ESM had both the highest annual precipitation (range
433 to 686 mm) and the highest trend in annual precipitation (range -2.1 to +0.8 mm yr$^{-1}$)
among the forcing data sets.
The spatial domain of interest is the Northern Hemisphere permafrost land regions. Our
analysis is based on the $0.5^o \times 0.5^o$ resolution gridded driving and modeled data for winter
(DJF) 1980-2000.
**2.2  Observations**
A data set of monthly near-surface air temperature, 20 cm soil temperatures and snow depth
from Russian meteorological stations have been provided by the All-Russian Research
Institute of Hydrometeorological Information-World Data Centre (RIHMI-WDC;
http://meteo.ru/) (Sherstyukov, 2008). 579 stations report snow depth and 268 stations provide
simultaneous data of all three variables. Ground surface temperature data are not available.
Precipitation station data have been compiled from the Global Summary of the Day (GSOD)
data set produced by the National Climatic Data Center (NCDC; http://www.ncdc.noaa.gov)
for all of the stations that are included in the RIHMI-WDC data set. We also use gridded
snow    water    equivalent    (SWE)    data    from    the    GlobSnow-2    product
(http://www.globsnow.info/swe/), which has been produced using a combination of passive
microwave radiometer and ground-based weather station data (Takala et al., 2011). Snow
depth was then calculated from SWE using a snow density of 250 kg m$^{-3}$. Orographic
complexity, vegetation cover, and snow state (e.g. wet snow) affect the accuracy of this
product. When compared with ground measurements, the GlobSnow product can show
regional differences (of ca. 0.5-5 cm) with biases increasing with increasing SWE (e.g.,
Takala et al., 2011; Muskett, 2012). All these data have been compiled for winter (DJF) and
the same time period of 1980-2000. This period was chosen because soil temperature data are
sparse before 1980 and the JULES simulation stopped in the year 2000. Comparison of the
simulations with the station data was done using a weighted bilinear interpolation from the 4
surrounding model grid points onto the station locations.



### 2.3 Analysis Methods

Our analysis is focused on the common winter (DJF) condition, although snow can begin in November and end at the beginning of May, but we checked that a different winter definition (NDJFMA) does not substantially change the results. The focus in our study is on the evaluation of the simulated air-soil temperature relationships, modulated by snow depth. For this, we analyze the winter mean as well as the interannual variability (expressed as the standard deviation) of 4 variables: near-surface air temperature ($T_{air}$), near-surface soil temperature (soil temperature at 20 cm depth; $T_{soil}$), snow depth ($d_{snow}$), and the difference between $T_{soil}$ and $T_{air}$. This difference $\Delta T$ ($\Delta T = T_{soil} - T_{air}$) is called the air-soil temperature difference. By limiting our analysis to the winter only, we are able to attribute the across-model and model-to-observation differences in $\Delta T$ primarily to snow insulation effects. We assume that there is relatively little impact due to soil moisture and texture between surface and 20 cm depth in winter. Although we recognize the difference between ground surface and 20 cm soil temperatures and that soil organic layer could play a role in certain locations (e.g., Romanovsky and Osterkamp, 1995), ground surface temperatures are not recorded in the Russian data set, while 20 cm soil depth temperatures are, hence our choice. However, we find that the results do not significantly change when the model simulated temperature differences between ground surface and near-surface air temperature are used instead of between 20 cm soil and near-surface air temperatures.

We use correlation analysis to investigate the co-variability between $\Delta T$ and $d_{snow}$ as well as between $T_{soil}$ and its two forcing factors ($T_{air}$ and $d_{snow}$). The input consists of detrended time series of winter means at each grid point. The calculated correlation maps (i.e. spatial distributions of correlation coefficients) based on model and observation data, allow the comparison of the spatial patterns of these relationships. Significance of correlation coefficients is estimated by the Student t-test.

To further examine the functional behavior between different variables, we present relation diagrams between pairs of variables (e.g. variation of $\Delta T$ with change of $d_{snow}$). To evaluate the performance of the individual LSMs we calculate the root-mean-square error (RMSE) between the observed and modeled relationships. We illustrate the dependence on air temperatures by evaluating the models and observations for several different near-surface air temperature ranges. We split the data into 3 regimes: the coldest conditions ($T_{air} \leq -25\ °C$) represent 24% of observations, the intermediate temperature conditions ($-25\ °C < T_{air} \leq -15\ °C$)




represent 42% of the observations, and the warmest conditions (-15 ℃ < $T_{air}$ ≤ -5 ℃) represent
34% of observations. The principal motivation for such classifications is to distinguish dry
snow pack regimes from those where sporadic melt may occur even in winter. Hence it is an
indirect separation of temperature-gradient metamorphosis regimes and density-gradient
metamorphosis snow pack regimes. Additionally, we present conditional probability density
functions (PDFs) of $\Delta T$ for different snow depth and air temperature regimes and compare the
simulated with those obtained from station observations.
## 3 Results
### 3.1 Relationship between air – soil temperature difference and snow depth
The air-soil temperature difference ($\Delta T$) - snow depth ($d_{snow}$) relationship in winter (Fig. 1)
shows that observations and all models produce a clear relationship between increase of $\Delta T$
and increases of $d_{snow}$. However, Fig. 1 also shows a wide across-model spread in the
simulated relationships, and that some of the models are not consistent with the behavior in
the observations. There is also significant scatter in the observation-based relationship, the
inter-quartile range of $\Delta T$ is 1.5-8.5 K at specific snow depth and air temperature regimes,
likely resulting from complicating factors such as snow pack density and moisture content
variability over the winter, as well as observational errors. Similar ranges of variability are
produced by several models (such as CLM4.5, CoLM and JULES), but other models (such as
ISBA and MIROC-ESM) produce noticeably smaller variations.
The Russian station data and some model results exhibit a linear relation between winter $\Delta T$
and $d_{snow}$ at relatively shallow snow depths with a trend towards asymptotic behavior at larger
snow depths (Fig. 1), which is in agreement with earlier findings (Zhang, 2005; Ge and Gong,
2010; Morse et al., 2011). However, only three models (CLM4.5, CoLM, JULES) reproduce
reasonably well the $\Delta T$ - $d_{snow}$ relationship seen in the observational station data (Fig. 1) using
a benchmark of RMSE < 5 K for all temperature regimes. In particular LPJ-GUESS,
ORCHIDEE, UVic, UW-VIC, MIROC-ESM show large RMSE for cold air conditions. ISBA
stands out overall, with a RMSE of 7-18 K in all temperature ranges. We conclude that these
models do not adequately represent the features of the observed $\Delta T$ /$d_{snow}$ relationship.
Figure 2 views the $\Delta T$ /$d_{snow}$ relationship in the complementary form of the PDFs of $\Delta T$ for
different snow depth and air temperature regimes. Since the Russian snow depths are clearly
non-Normal in distribution (SI Fig. 1, with a median $d_{snow}$ of 30 cm), we divide the data into
"shallow" ($d_{snow} \leq 20$ cm) and "thick" ($d_{snow} \geq 45$cm) regimes. The modal value of the station
data $\Delta T$ PDF is 5 K for "shallow" snow and 14 K for "thick" snow - that is thick snow is a
better insulator than thin snow. Based on the $\Delta T$ PDFs, five models (CoLM, CLM4.5, JULES,
ORCHIDEE, MIROC-ESM) successfully separate the $\Delta T$ regimes under different snow depth
conditions, while the other models clearly fail for at least one of these snow depth regimes.
However, even for the better models, both the shapes and the modal values of the simulated
PDFs differ from the observed PDF.
Both Figs. 1 and 2 indicate that air-soil temperature differences are related to air temperature
conditions. This is due to snow pack properties, particularly its density and moisture content,
that affect the air-soil temperature difference. For example, the density of fresh fallen snow
tends to be much lower under cold air temperatures than warm (Anderson, 1976), leading to
increased insulation (larger $\Delta T$). Snow densification is also a function of air temperature, for
example, depth hoar metamorphosis of the snow pack, which produces more insulation
(loosely packed depth-hoar crystals have very low thermal conductivity), is promoted by
strong thermal gradients in the snow pack, and is typical of continental climates (e.g., Zhang
et al., 1996).
The observations in Figs. 1 and 2 indicate that snow under colder climates have greater
insulation than under warmer climates. This is shown by a larger $\Delta T$ for colder $T_{air}$ than for
warmer $T_{air}$ (for a certain snow depth) and a greater sensitivity of $\Delta T$ to changes in $d_{snow}$ (Fig.
1), and by the larger modal value of the $\Delta T$ PDF for colder $T_{air}$ than for warmer $T_{air}$ (21 K for
$T_{air} \leq$ -25 °C and 9 K for -15 °C $< T_{air} \leq$ -5 °C; Fig. 2). This is consistent with colder climates
having lower density snow packs, and the differences are in line with measurements of snow
density variability (Zhong et al., 2013). Additionally, both the inter-quartile range in Fig. 1
and the width of the PDFs in Fig. 2 become larger as air temperatures cool. This may be
related to the formation of depth hoar, which is a very good insulator and its varying presence
in the snow pack decouples $\Delta T$ from $d_{snow}$. Cold, thin snow packs tend to contain much more
low density depth hoar than warmer snow packs (e.g., Zhang et al., 1996; Singh et al., 2011).
Continental regions have large annual temperature cycles, with greater interannual variability
and thinner snow packs, than maritime ones. This variability leads to greater scatter and
greater sensitivity of the $\Delta T / d_{snow}$ relationship in the cold winter regions. An additional cause
of scatter is that the density of fresh-fallen snow decreases with falling temperature.
Accordingly, we find in the cold air temperature regime ($T_{air} \leq$ -25 °C) a larger $\Delta T$ in early





winter (November-December) when the snow pack is composed of thin, low density fresh
snow (and depth hoar) than in late winter (January-February) (SI Fig. 2). Under warm
conditions (-15 °C < $T_{air}$ ≤ -5 °C) such a separation is not observed.
Our analysis (Fig. 1) indicates that some models (CLM4.5, CoLM, JULES) are better able to
replicate the observed effect of air temperature on the $\Delta T$ /$d_{snow}$ relationship than others (LPJ-
GUESS, MIROC-ESM, ORCHIDEE, UW-VIC). The latter do not fully replicate the larger
$\Delta T$ under cold air temperature conditions. CLM4.5, CoLM and JULES capture a larger $\Delta T$ for
colder air temperatures for a given $d_{snow}$ in agreement with the observations. However, for
shallow snow JULES simulates twice as large increase of $\Delta T$ with increasing $d_{snow}$ for all
temperature ranges, compared with observations. Two models (ISBA, UVic) clearly fail in
this evaluation. Poor model performance in reflecting air temperature influence on the $\Delta T$
/$d_{snow}$ also manifests itself in regime separation of the PDFs (Fig. 2). Some models do not
separate the $\Delta T$ regimes under different air temperature conditions well or at all (ISBA, LPJ-
GUESS, MIROC-ESM, UVic), while others cannot capture the observed cold temperature
regime features (i.e., too broad PDFs and shifts towards smaller modal values; ORCHIDEE,
UW-VIC). The three models with reasonable inter-variable relations (CLM4.5, CoLM,
JULES) also capture the regime separation in the PDFs. These three models as well as LPJ-
GUESS and ORCHIDEE also represent the observed greater insulation of early winter snow
packs under cold conditions (SI Fig. 2).
The maps of the $\Delta T$ /$d_{snow}$ correlations in winter (Fig. 3) demonstrate the strong spatial
variability in the $\Delta T$ /$d_{snow}$ relationship, but indicate that most models agree on the general
large-scale pattern. Some models (CLM4.5, CoLM, ORCHIDEE, UW-VIC) show a
reasonable pattern correlation coefficient (r ≥ 0.4) with observations, while the others do not.
Most models simulate the highest positive correlation in the region of the East Siberian Plain
and Siberian High lands. In some regions, namely in Scandinavia, West Russian Arctic, West
and Central Siberian Plains, the correlation is much weaker and often not statistically
significant. These are the regions of large winter snow depth (Sect. 4.1.2) which are
influenced by North Atlantic cyclonic activity which brings relatively warm moist air and
heavy precipitation in winter (and a positive correlation between snow depth and air
temperature), leading to relatively small mean $\Delta T$. Obvious outliers in the $\Delta T$ /$d_{snow}$ correlation
map are the LPJ-GUESS and UVic models, which do not reproduce the observed pattern of
correlation. UVic calculates a reverse pattern correlation than observations for many regions



(e.g. significant positive correlation in West Siberian Plain and Central Siberian Highlands).
LPJ-GUESS produces very few statistically significant correlations. The model correlations
are likely highly sensitive to the quality of the snowfall forcing data, which is uncertain across
much of the region due to limited station data that go into most global snowfall products
(Hancock et al., 2014; Drobot et al., 2006).
**3. 2   Variability of soil temperature with air temperature and snow depth**
Next we assess whether or not the models can correctly reproduce the interannual near-
surface soil temperature ($T_{soil}$) variability in relation to snow depth ($d_{snow}$) and near-surface air
temperature ($T_{air}$) variability. Previous authors (Smith and Riseborough, 2002; Sokratov and
Barry, 2002; Zhang, 2005; Lawrence and Slater, 2010) have noted that the strength of
relationship between $T_{soil}$ and $T_{air}$ is modulated by $d_{snow}$ and the snow insulation effect
increases only up to a limiting depth beyond which extra snow makes little difference to soil
temperatures. Zhang (2005) reports that the limiting snow depth is approximately 40 cm.
To inspect the difference of the insulation effects on both sides of such a limiting snow depth,
we investigate the $T_{soil}/T_{air}$ relationship under shallow ($d_{snow} \leq 20$ cm) and thick ($d_{snow} \geq 45$ cm)
snow conditions. Observations showed that the slope of this relationship is higher when the
snow cover is thin, compared with thicker snow conditions (e.g., for Yukon Territory in
Canada; Karunaratne and Burn, 2003). Indeed, the Russian observations (Fig. 4, Table 2)
indicate a three times stronger $T_{soil}/T_{air}$ relationship (0.62 ℃/ ℃, $R^2$=0.8) under shallow snow
pack than thicker snow conditions (0.21 ℃/ ℃, $R^2$=0.4). The average $d_{snow}$ in the shallow snow
regime is 13.7 cm and that for the thick snow regime is 58.5 cm, so we would expect, if near-
surface air temperature and conductivities were equal in both snow depth classes, a ratio of
4.3 in the slopes of Fig. 4. The models that better reproduce the observed $\Delta T /d_{snow}$
relationship (CLM4.5, CoLM, JULES) reproduce the observed variation in the $T_{soil}/T_{air}$
relation better than others. JULES and CoLM indicate a factor of 4 change, while CLM4.5
indicates a factor of 2 change. Other models (LPJ-GUESS, MIROC-ESM, ORCHIDEE)
strongly underestimate the increase of the slope for decreasing snow depth (they simulate a
factor change of about 1.5). The two models that had unrealistic $\Delta T /d_{snow}$ relationships (ISBA,
UVic) also fail in this evaluation of their $T_{soil}/T_{air}$ relationship. They simulate too strong $T_{soil}$
$/T_{air}$ relationships (gradients larger than 0.9 ℃/ ℃, $R^2$>0.7; Table 2) that are almost completely
independent of the snow depth regimes, particularly in ISBA, which is not consistent with
observations. These models' spatial correlation patterns between $T_{soil}$ and $T_{air}$ also differ
greatly from the observations and the other models (SI Fig. 3) and show very high positive



correlation (r > 0.8) in most regions, as may be expected from the strong relationship shown
in Fig. 4. The RMSE of their modeled $T_{soil}/T_{air}$ relationships from observations reaches ca.
10 ℃ .
The $T_{soil}/d_{snow}$ relationship (Fig. 5) displays the variation of $T_{soil}$ with changing snow depth
and emphasizes the weakening role of snow depth for $T_{soil}$ under thick snow conditions. With
increasing $d_{snow,}$ soil temperatures asymptotically converge towards a value of around 0 ℃.
Overall, the Russian observations indicate that snow depth above about 80-90 cm has very
little additional insulation effect on $T_{soil}$. Most of the models show consistent results with
regard to this aspect, although the inter-quartile range of $T_{soil}$ for a certain snow depth  is quite
large in some models (ISBA, ORCHIDEE, UVic, UW-VIC) (Fig. 5). The figure further points
to the air temperature dependency of the relation. On average, for a given $d_{snow}$, a colder $T_{soil}$
is observed for colder near-surface air temperatures, compared with warmer air temperatures.
Most models can replicate this effect of $T_{air}$ on the $T_{soil}/d_{snow}$ relationship, though with
differing accuracy. The RMSE between the observed and modeled relationships can reach ca.
10 ℃ and more (in ISBA, UVic, UW-VIC), particularly under cold conditions.
The spatial patterns of the correlation coefficients between $T_{soil}$ and $T_{air}$ (SI Fig. 3) and
between $T_{soil}$ and $d_{snow}$ (SI Fig. 4) show a relatively large across-model scatter in the specific
regions. Obvious outliers in the $T_{soil}/T_{air}$ correlation maps (SI Fig. 3) are ISBA and UVic
which strongly overestimate the correlation (r > 0.9) over most of the Arctic. Other models
(LPJ-GUESS, ORCHIDEE, UW-VIC) also overestimate the correlation in some regions (e.g.
western Russian Arctic, r > 0.7). Most of the simulated maps of $T_{soil}/d_{snow}$ correlation (SI Fig.
4) agree with the observations on a strong positive correlation in East Siberia. This is a region
of relatively shallow snow (10-40 cm; Fig. 6) and there $T_{soil}$ is very sensitive to variations in
snow depth (e.g., Romanovsky et al., 2007). Comparing both simulated correlation maps, it is
obvious that in this region, $T_{soil}$ correlates more strongly with $d_{snow}$ than with $T_{air}$, in agreement
with the Russian data and earlier studies (Romanovsky et al., 2007; Sherstyukov, 2008).
**4   Roles of atmospheric forcing and model processes**
The across-model differences in the snow insulation effect, presented by the air temperature -
snow depth - soil temperature relationships described above, are partially due to differences in
the atmospheric forcing data and also due to differences in the snow and soil physics used in
the LSMs. However, because the climate forcing data sets utilized with each model are





observation-based (except for MIROC-ESM), obvious outliers in individual model
performance likely mainly indicate poor or deficient physical descriptions of the air/snow/soil
relations in that specific LSM.

### 4.1 Atmospheric forcing and snow depth

#### 4.1.1 Air temperature and precipitation

Both near-surface air temperature and precipitation are given by the climate forcing data sets
(SI Table 1) for all models, except for MIROC-ESM which simulates both. The across-model
differences in forcing air temperature used are relatively small and the simulated spatial
patterns of temperature are very similar (SI Fig. 5). All forcing datasets are somewhat colder
than Russian station data in their grid cells. The biases range from -0.8 K to -4.7 K (SI Table
2), reflecting biases in the climate forcing data used by the models. In contrast, MIROC-ESM
has a positive (mean) air temperatures bias of +2.7 K.
The large-scale patterns of precipitation are similar across the models, but regional differences
can be large (SI Fig. 6). The individual differences in precipitation range from  -0.2 mm/day
to +0.5 mm/day (SI Table 2) relative to the average of the Russian station data. Unfortunately,
snowfall was archived in only a few models, however large-scale spatial patterns are similar
across these models (SI Fig. 7).

#### 4.1.2 Snow depth

The broad-scale spatial snow depth ($d_{snow}$) patterns are similar across the models and show
general agreement with the observed patterns (Fig. 6). The well-pronounced areas of
maximum winter $d_{snow}$ (50-100 cm) are in Scandinavia, the Urals, the West Siberian Plain,
Central Siberian Highlands, the Far East, Alaskan Rocky mountains, and Labrador Peninsula
and isle of Newfoundland. However, large regional across-model variability is obvious. Some
models (JULES, LPJ-GUESS, ORCHIDEE, UVic) underestimate $d_{snow}$, while others
(CLM4.5, CoLM, ISBA, UW-VIC) overestimate it (Fig. 6; SI Table 3). The model biases are
quite similar with respect to station observations and GlobSnow data. The evaluation of the
model performance for SWE compared to GlobSnow indicates the same bias characteristics
as described here for snow depth (not shown). It should be noted, that the models do not
account for snowdrift. However, redistribution of snow due to wind is an important aspect,
which makes comparison between in-situ measured and modeled snow depths difficult (e.g.,
Vionnet et al., 2013; Sturm and Stuefer, 2013; Gisnas et al., 2014).



Precipitation/snowfall across-model differences cannot be the primary explanation of these $d_{snow}$ differences since some models (JULES, MIROC-ESM, ORCHIDEE) have positive bias in precipitation (> 0.2 mm/d, SI Table 2) but simulate much lower $d_{snow}$ compared to other models (Fig. 6, SI Figs. 6, 7, SI Table 3). Across-model differences in the interannual variability of winter precipitation do not translate simply to corresponding differences in the interannual $d_{snow}$ variability (not shown). For example, UVic calculates the (unrealistically) largest interannual $d_{snow}$ variability in the boreal Europe permafrost region which is not reflected in the precipitation variability. These results indicate that the simulated snow depth is a function of both, the prescribed winter precipitation, and the model's snow energy and water balance.

## 4.2  Model processes

We have shown that the across-model spread in the representation of snow insulation effects (Sects. 3.1, 3.2) can not predominantly be explained by differences in the forcing data (Sect. 4.1), but to a large extent is due to the representation of snow processes in the models. By considering the relationship plots and the conditional PDFs (Figs. 1, 2, 4, and 5) we were able to sort the models in terms of their snow insulation performance. In this section we discuss the influence of the different snow parameterizations in the models.

Models with better performance (CLM4.5, CoLM, JULES) apply multi-layer snow schemes. This allows them to simulate more realistic (stronger) insulation because they consider the snowpack's vertical structure and variability. They calculate the energy and mass balance in each snow layer, are able to capture nonlinear profiles of snow temperature, and can also account for thermal insulation within the snowpack such as when the upper layer thermally insulates the lower layers (e.g., Dutra et al., 2012). These models also incorporate storage and refreezing of liquid water within the snow, parameterize wet snow metamorphism, snow compaction, and snow thermal conductivity (Table 1), which have been found to be among the most important processes for good snow depth and surface soil temperature simulation (e.g., Wang et al., 2013).

An underestimated snow depth directly leads to insulation that is too weak in JULES, LPJ-GUESS, ORCHIDEE, and UVic (Fig. 6, SI Table 3). However only in ORCHIDEE and UVic does this lead to a significant underestimation of $\Delta T$ (SI Table 3, SI Fig. 8) indicating bias



compensation in the two other models. Thus, compensating error effects occur due to snow
density and conductivity (SI Fig. 9, Table 1), which impact snow thermal insulation.
Our analysis showed that two models (ISBA, UVic) have $T_{soil}/T_{air}$ correlation that are too
high indicating that they do not represent the modulation of the $T_{soil}/T_{air}$ relationship by snow
depth (Fig. 4). This is consistent with their underestimation of $\Delta T$ (Figs. 1 and 2, SI Fig. 8, SI
Table 3). In UVic, the snowpack is treated not as a separate layer but as an extension of the
top soil layer and a combined surface-to-soil thermal conductivity is calculated (Table 1).
Such a scheme largely negates or reduces the insulating capacity of snow (Slater et al., 2001).
Koven et al. (2013) noted that such a scheme simulates very little warming of soil, and
sometimes even cooling. The slightly underestimated snow depth (SI Table 3, Fig. 6)
contributes (but not as the primary factor) to reduced snow insulation, as reported for UVic
(Avis, 2012).
ISBA strongly underestimates $\Delta T$, while strongly overestimating $d_{snow}$, compared with
observations (SI Table 3, Fig. 6). However, ISBA uses the same atmospheric forcing data as
JULES (accordingly the air temperature and precipitation are quite similar; SI Table 2). Also,
the model's snow density (150-250 kg m$^{-3}$) is similar to other models (CLM45, CoLM,
JULES) (SI Fig. 9) and in agreement with Zhong et al. (2013) who report snow density values
of on 180-250 kg m$^{-3}$ for tundra/taiga and 156-193 kg m$^{-3}$ for alpine snow classes in winter.
This apparent contradiction comes from the parameterization of snow cover fraction within
each grid cell (SCF). The version of ISBA used here calculates a unique superficial soil
temperature whether or not the soil is covered by snow and all the energy and radiative fluxes
are area-weighted by SCF (equations 7 and 20 in *Douville et al.*, 1995). In order to get
reasonable albedos in snow-covered forests, as is necessary when ISBA is coupled to the
CNRM-CM climate model, the parameterization gives very low SCF in the boreal forest
(between 0.2 and 0.5). Hence, snow insulates only 20% to 50% of the grid cell, despite fairly
high snow depths. The heat fluxes from the snow-covered fraction are averaged with the
fluxes from the snow-free surface, strongly concealing the actual insulating effect of snow
and underestimating it over the grid cell. Using the detailed snow model Crocus (Brun et al.,
1992; Vionnet et al., 2012) with a SCF equal to 100% leads to an almost perfect simulation of
near-surface soil temperature over Northern Eurasia (Brun et al., 2013). A similar experiment
with ISBA and a SCF equal to 100% (Decharme et al., 2015) leads to good performances
showing that the low $\Delta T$ in ISBA despite high snow depth in the present study is mostly due





to this sub-grid snow fraction. The results are further improved by updating the snow albedo
and snow densification parameterization.
Interestingly, the ORCHIDEE performance in simulating snow depth and $\Delta T$ is similar to
UVic (underestimation of $d_{snow}$ and $\Delta T$; SI Table 3). However, ORCHIDEE can better
represent the observed $T_{soil}/T_{air}$ relationship and its modulation due to snow pack.
ORCHIDEE employs, similarly to UVic, a fixed snow density and thermal conductivity.
However, in contrast with UVic, ORCHIDEE applies a multi-layer scheme and simulates heat
diffusion in the snowpack in up to 7 discrete layers (Table 1; Koven et al., 2009). This helps
resolving the snow thermal gradients between the top and the base of the snow cover, and
might explain how some of the snow insulation effects are reasonably represented in
ORCHIDEE, despite the simpler treatment of temperature diffusion.
**5  Summary and conclusions**
The aim of this work was to evaluate how state-of-the-art LSMs capture the observed
relationship between winter near-surface soil and air temperatures ($T_{soil}$, $T_{air}$) and their
modulation by snow depth ($d_{snow}$) and climate regime. We presented some benchmarks to
evaluate model performance. The results are based on the comparison of LSMs with a novel
comprehensive Russian station data set.
We see large differences across the models in their mean air-soil temperature difference ($\Delta T$)
of 3 to 14 K, in the gradient between near-surface soil and air temperatures ($T_{soil}/T_{air}$) (0.49 to
0.96 ℃/℃ for shallow snow, 0.13 to 0.93 ℃/℃ for thick snow), and in the increase of $\Delta T$
with increasing snow depth (modal value of $\Delta T$ PDF: 0 to 10 K for shallow snow, 5 to 21 K
for thick snow). Most of the nine models compare to the observations reasonably well
(observations: $\Delta T = 12$ K, modal $\Delta T$ values of 5 K for shallow snow and of 14 K for thick
snow, $T_{soil}/T_{air} = 0.62$ ℃/℃ for shallow snow, $T_{soil}/T_{air} = 0.21$ ℃/℃ for thick snow). Several
models also capture the modulation by air temperature condition (larger increase in $\Delta T$ with
increasing $d_{snow}$ under colder conditions) and display the control of snow depth on $T_{soil}$
(weaker $T_{soil}/T_{air}$ relationship under thicker snow).  However, while they generally capture
these observed relationships, their strength can differ in the individual models. Two models
(ISBA, UVic) show the largest deficits in snow insulation effects and cannot separate the $\Delta T$
regimes neither for different snow depths nor for different air temperature conditions.





The primary aim of this study was to use this ensemble of models to document model
performance with respect to $T_{soil}$ versus $T_{air}$ relationships, and to identify those with better
performance, rather than to quantify the best model. We were able to attribute performance
strength/weakness to snow model features and complexity. Models with better performance
apply multi-layer snow schemes and consider complex snow processes (e.g. storage and
refreezing of liquid water within the snow, wet snow metamorphism, snow compaction).
Those models which show limited skill in snow insulation representation (underestimated $\Delta T$,
very weak dependency of $\Delta T$ on $d_{snow}$, almost unity ratio of $T_{soil}/T_{air}$) have some deficiencies
in the simulation of heat transfer in snow and soil layer, particularly in the representation of
snow depth and density (conductivity). We also emphasize that compensating errors in snow
depth and conductivity can occur. For example, an excessive correlation between $T_{soil}$ and $T_{air}$
can be attributed to excessively high thermal conductivity even when the snow depth is
correctly (or over) simulated. This finding underscores the need for detailed model
evaluations using multiple, independent performance metrics to establish that the models get
the right functionality for the right reason. It should be noted that the treatment of ground
properties, particularly soil organic matter and soil moisture/ice content, also affect the
simulated winter ground temperatures. The specific evaluation of these individual processes is
more robustly investigated with experiments conducted for individual models (e.g. recently,
Wang et al., 2013; Gubler et al., 2013; Decharme et al., 2015).
A realistic simulation of the snow is a key pre-requisite for accurate modeling of the soil
thermal dynamics across the permafrost region. The areal cover of Northern Hemisphere near-
surface permafrost varies greatly across the nine models in the hindcast simulation (1960-
2009). Some of the better performing snow insulation effect models (CLM4.5, JULES)
simulate a near-surface permafrost area of 12 to 16 million km$^2$, which is comparable with the
IPA map estimate (16 million km$^2$) (Brown et al., 1997; Slater and Lawrence, 2013). CoLM
and ORCHIDEE, identified as reasonable models with respect to snow insulation, simulate
much lower (7 million km$^2$) and higher (20 million km$^2$) areas, respectively. However, ISBA,
one of the two models that showed rather limited skill in representing snow insulation effects,
also simulates the highest permafrost area (20 million km$^2$). This is consistent with previous
studies (e.g., Vavrus, 2007; Koven et al., 2013) which concluded that first-order control on
modeled near-surface permafrost distribution is the representation of the air-to-surface soil
temperature difference. When the models with poor snow models are eliminated, there is no
clear relationship between the quality of the snow insulation in the models and the simulated



area of permafrost, likely because several other factors such as differences in the treatment of soil organic matter, soil hydrology, surface energy calculations, model soil column depth, and vegetation also provide important controls on simulated permafrost distribution (Marchenko and Etzelmüller, 2013).

*Acknowledgments.* The data will be made available through the National Snow and Ice Data Center (NSIDC; http://nsidc.org); the contact person is Kevin Schaefer (kevin.schaefer@nsidc.org). This study was supported by the Permafrost Carbon Vulnerability Research Coordination Network, which is funded by the U.S. National Science Foundation (NSF). Any use of trade, firm, or product names is for descriptive purposes only and does not imply endorsement by the U.S. Government. E.J.B. was supported by the Joint UK DECC / Defra Met Office Hadley Centre Climate Program (GA01101). E.J.B., S.P., P.C. and G.K. were supported by the European Union Seventh Framework Program (FP7/2007-2013) under grant agreement n°282700. T.J.B. was supported by grant 1216037 from the NSF Science, Engineering and Education for Sustainability (SEES) Post-Doctoral Fellowship program. This research was sponsored by the Integrated approaches and impacts, China Global Change Program (973 Project), National Basic Research Program of China Grant 2015CB953602 and the National Natural Science Foundation of China Grant 40905047.

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



**Tables**
**Table 1.** PCN snow model details.

| Model Reference for snow scheme | Snow scheme[1] | Snow layers | Water phases | Liquid water treatment[2] | Snow density[3] | Snow thermal conductivity[4] |
|---|---|---|---|---|---|---|
| CLM4.5 Swenson and Lawrence, 2012 Oleson et al., 2013 | ML | Dynamic (max. 5) | Liquid, Ice | Bucket-type prognostic in each layer | depends on snow depth; compaction 3) a,b,c | quadratic equation on ρ |
| CoLM Dai et al., 2003 Ji et al. 2014 | ML | Dynamic (max. 5) | Liquid, Ice | Bucket-type prognostic in each layer | depends on snow depth; compaction 3) a,b,c | quadratic equation on ρ |
| ISBA Boone and Etchevers, 2001 | ML | Static (3) | Liquid, Ice, Vapor | Diagnosed from snow temperature, mass, density | compaction 3) a,b | quadratic equation on ρ, contribution due to vapor transfer |
| JULES Best et al., 2011 | ML | Dynamic (max. 3) | Liquid, Ice, Vapor | Bucket-type prognostic in each layer | compaction 3) a | power equation on ρ |
| LPJ-GUESS Gerten et al., 2004 Wania et al., 2009 | BL | Static (1) | Ice | Not represented | fixed 362 kg m$^{-3}$ | fixed 0.196 Wm$^{-1}$K$^{-1}$ |
| MIROC-ESM Takata et al., 2003 | ML | Dynamic (max. 3) | Ice | Not represented | fixed 300 kg m$^{-3}$ | fixed 0.3 Wm$^{-1}$K$^{-1}$ |
| ORCHIDEE Gouttevin et al.,2012 | ML | Dynamic (max. 7) | Ice | Not represented | fixed 330 kg m$^{-3}$ | fixed 0.25 Wm$^{-1}$K$^{-1}$ for tundra, 0.042 Wm$^{-1}$K$^{-1}$ for taiga |
| UVic Meissner et al., 2003 Avis, 2012 | I | Static (1) | Ice | Not represented | fixed 330 kg m$^{-3}$ | bulk conductivity |
| UW-VIC Andreadis et al., 2009 | BL | Dynamic (max. 2) | Liquid, Ice, Vapor | Constant liquid water holding capacity | compaction 3) a,b | fixed 0.7 Wm$^{-1}$K$^{-1}$ |

[1] ML: Multi-layer, BL: Bulk-layer, I: Implicit; according to Slater et al. (2001)
[2] Not represented means dry snow
[3] Processes for densification of the snow: a) mechanical compaction (due to the weight of the overburden), b)
thermal metamorphosis (via the melting–refreezing process), c) destructive metamorphism (crystal breakdown
due to wind, thermodynamic stress); Anderson (1976), Jordan (1991), Kojima (1967)
[4] quadratic equation on ρ according to Jordan (1991), Anderson (1976); contribution due to vapor transfer
according to Sun et al.(1999)



**Table 2.** Sensitivity of near-surface soil temperature ($T_{soil}$) to air temperature ($T_{air}$) in winter
(DJF) calculated by the slopes of the linear regression between $T_{soil}$ (℃) and $T_{air}$ (℃) for
different regimes of snow depth ($d_{snow}$), using data from all Russian station grid points and 21
individual winter 1980-2000. All relationships are statistically significant at p ≤ 0.01.

| | Snow depth regimes | | | |
| | Shallow | | Thick | |
| | $d_{snow} \leq 20$ cm | | $d_{snow} \geq 45$ cm | |
| | $T_{soil}/T_{air}$ (℃/℃) | $R^2$ | $T_{soil}/T_{air}$ (℃/℃) | $R^2$ |
|---|---|---|---|---|
| Observation | 0.62 | 0.79 | 0.21 | 0.41 |
| CLM4.5 | 0.69 | 0.89 | 0.33 | 0.56 |
| CoLM | 0.49 | 0.73 | 0.13 | 0.44 |
| ISBA | 0.93 | 0.98 | 0.93 | 0.94 |
| JULES | 0.68 | 0.77 | 0.19 | 0.46 |
| LPJ-GUESS | 0.73 | 0.89 | 0.52 | 0.75 |
| MIROC-ESM | 0.78 | 0.98 | 0.49 | 0.67 |
| ORCHIDEE | 0.86 | 0.83 | 0.56 | 0.64 |
| UVic | 0.96 | 0.97 | 0.81 | 0.68 |
| UW-VIC | 0.54 | 0.74 | 0.76 | 0.65 |



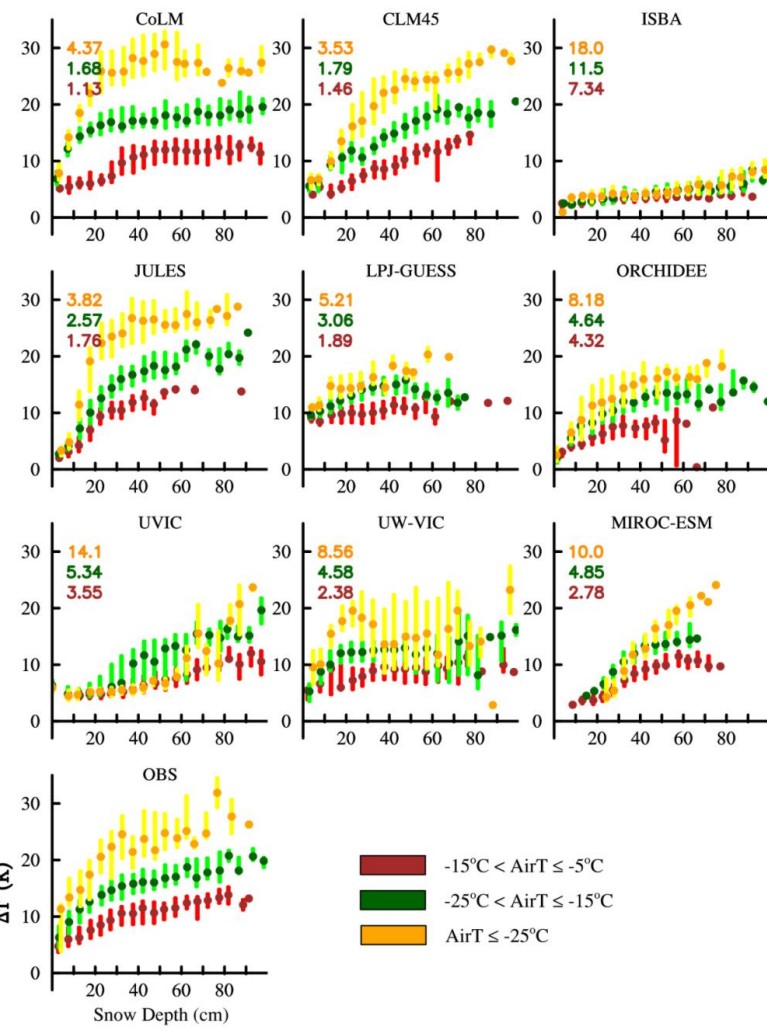

**Figure 1.** Variation of ΔT (K), the difference between soil temperature at 20 cm depth and air
temperature) with snow depth (cm) for winter 1980-2000. The dots represent the medians of 5
cm snow depth bins and the upper and lower bars indicate the 25[th] and 75[th] percentiles,
calculated from all Russian station grid points (n=268) and 21 individual winters. The
numbers in each model panel indicate the RMSE between the observed and modeled
relationship. Color represents different air temperature regimes.




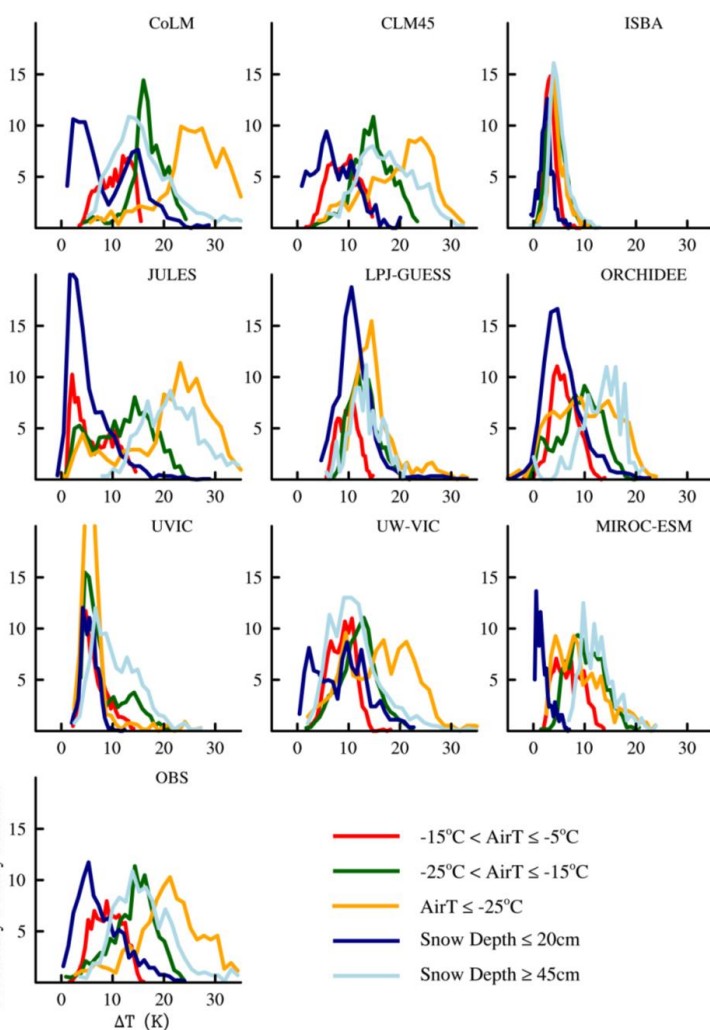

**Figure 2.** Conditional probability density functions (PDFs) of ΔT (K), the difference between soil temperature at 20 cm depth and air temperature for different snow depth and air temperature regimes (color) for winter 1980-2000.





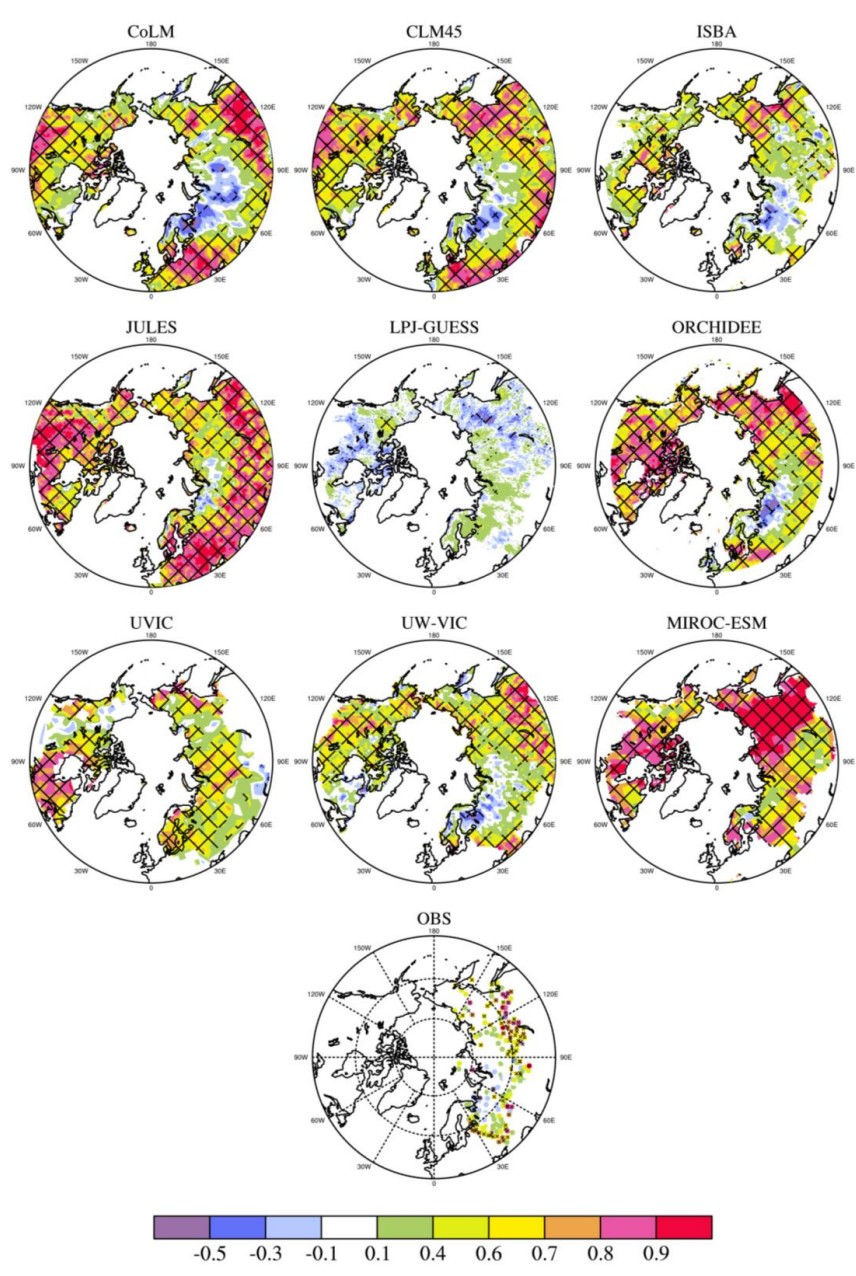

**Figure 3.** Spatial maps of the correlation coefficients between snow depth and ΔT, the

difference between soil temperature at 20 cm depth and air temperature for winter 1980-2000.

Regions with greater than 95% significance are hashed.



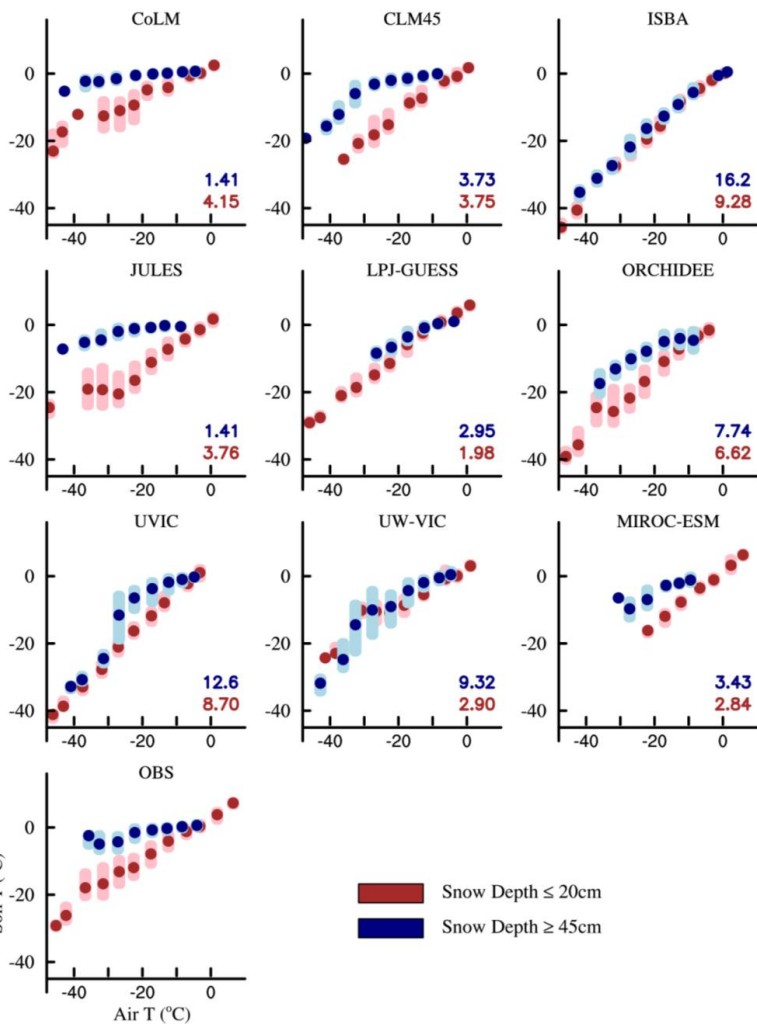

**Figure 4.** Variation of soil temperature at 20 cm depth (℃) with air temperature (℃) for
winter 1980-2000. The dots represent the medians of 5℃ air temperature bins and the upper
and lower bars indicate the 25[th] and 75[th] percentiles, calculated from all Russian station grid
points (n=268) and 21 individual winters. The numbers in each model panel indicate the
RMSE between the observed and modeled relationship. Color represents different snow depth
regimes.



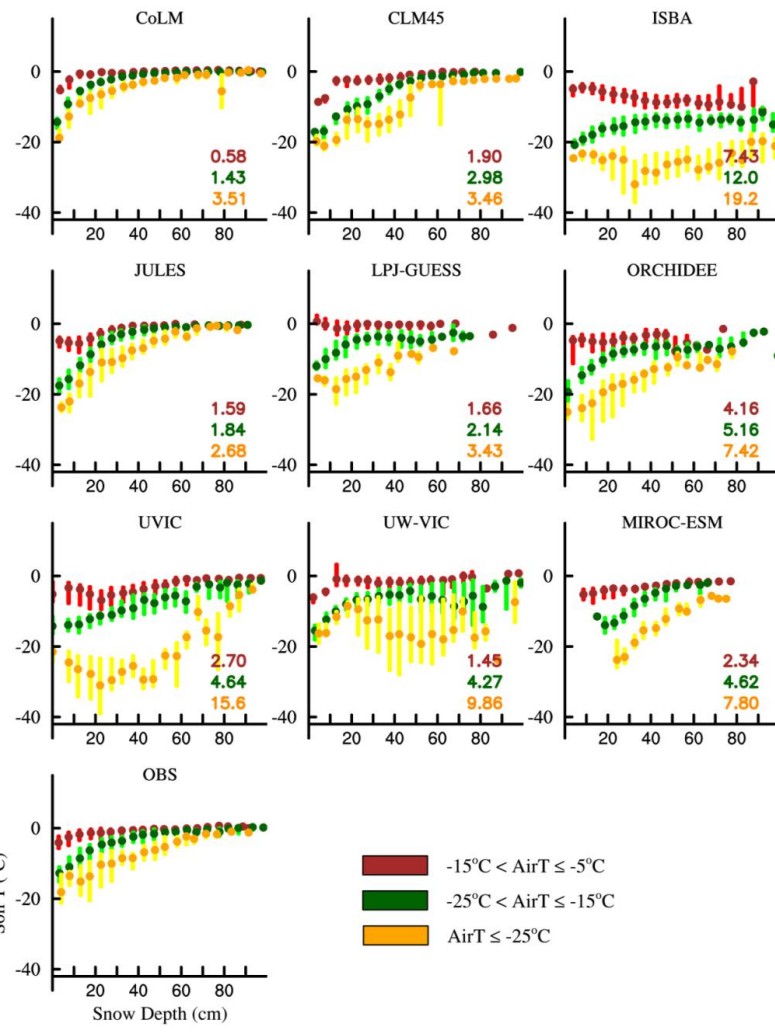

2
**Figure 5.** Variation of soil temperature at 20 cm depth (°C; y axis) with snow depth (cm) for
winter 1980-2000. The dots represent the medians of 5 cm snow depth bins and the upper and
lower bars indicate the 25[th] and 75[th] percentiles, calculated from all Russian station grid points
(n=268) and 21 individual winters. The numbers in each model panel indicate the RMSE
between the observed and modeled relationship. Color represents different air temperature
regimes.



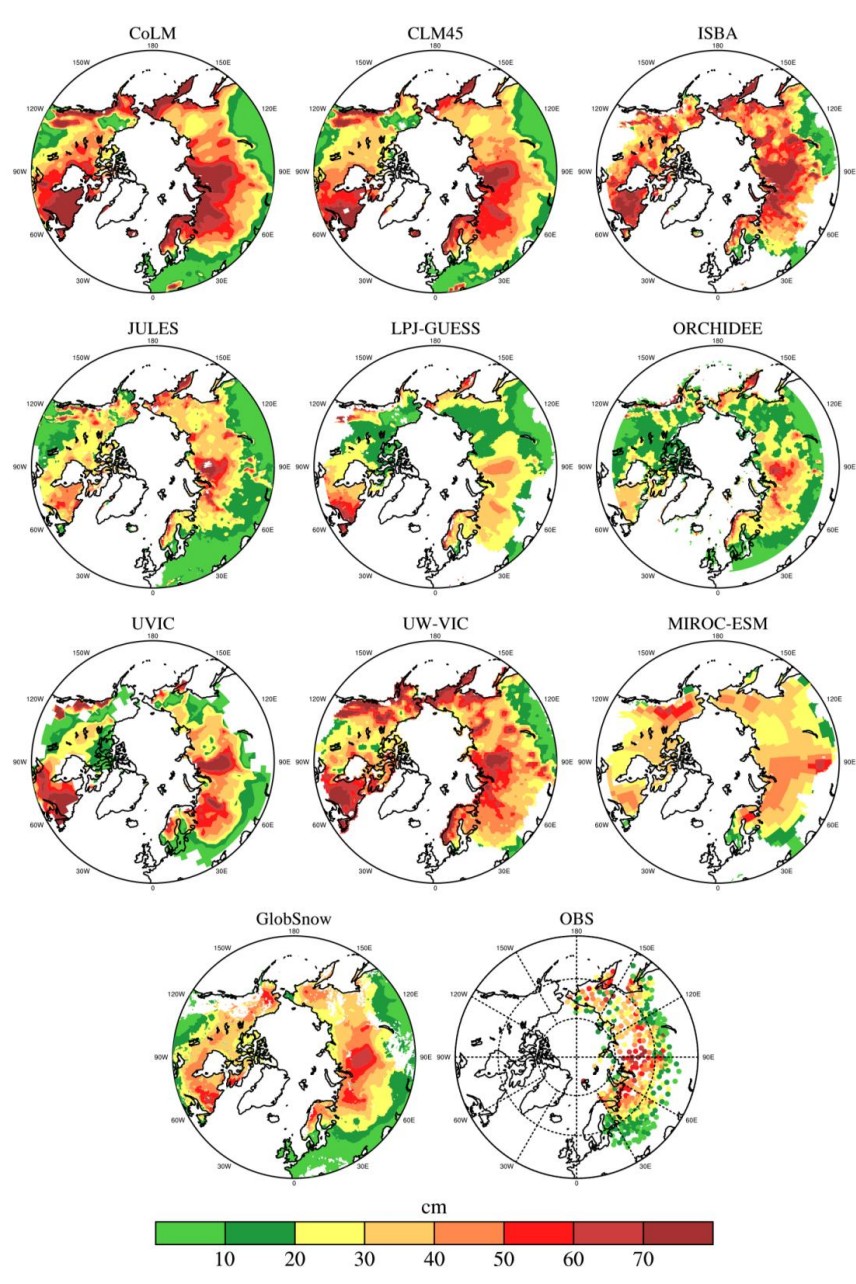

3    **Figure 6.** Spatial maps of snow depth (cm) for winter 1980-2000.