# Peer review of "Evaluation of air-soil temperature relationships"

_The Cryosphere, 2016_

## Referee Comment (RC1) · Anonymous Referee #1 · 31 Mar 2016

**General comments**

Snow and its insulation effects are critical for accurately simulating soil temperature and permafrost in high latitudes. This paper assessed the skills of nine land surface models based on the response patterns of $T_{soil}$ and the difference of $T_{soil}$-$T_{air}$ to snow depth in winter in high latitudes. The observed patterns at 268 climate stations in Russia were used as ground truth. Such an assessment is better than direct point-by-point comparison with station observations. It reveals some structural issues of the models in simulating snow depths and its insulation effects on soil temperature. The results from the observation stations are interesting as well. The data source is solid, the results and analysis are detailed and well presented in most parts. It is worthy to be published.

 **Specific comments**

The authors put several lines in abstract about near-surface permafrost. However permafrost results were not described in results and discussion sections, and it is only mentioned in summary and conclusion section. A somewhat proportional amount of description (in terms of length or importance) should be given in results and discussion sections so that it can be included in the summary and abstract. You need to add at least one paragraph about permafrost in the result (as suggested below) or in discussion sections.

P.9: Before analyzing the $T_{air}$-$d_{snow}$-$T_{soil}$ relationship, it would be interesting to briefly describe the modeled distribution and errors in snow depth and soil temperature comparing with observations in Russia. The section 4.1.2 about snow depth can be moved to here (table 3 in supplementary can be moved to here as well), and add something similar about the soil temperature. With the soil temperature results, you may add the results of permafrost extent and distribution as you mentioned in the summary and abstract. You may well aware and it is worthwhile to emphasize that the simulated snow depth and soil temperature could be influenced by inputs of the model, and the station observations have limitations in spatial coverage (covers only part of Russia, and may not well represent the grids). However, the response patterns of $T_{soil}$ and $T_{soil}$- $T_{air}$ To $D_{snow}$ should be consistent and can real deeper structural issues of the models.

P.8, Line 11-13: "We assume that …in winter". I feel such an assumption is not necessary. The effects of soil moisture and texture do have effects but is much smaller than that of snow. You may revise it to "The effects of other factors on ΔT are much smaller than that of snow" or delete the sentence.

P12, L2-5: This sentence does not connect well with the previous one (why LPJ-GUESS produces very low correlation coefficients). In addition, the meaning of the sentence is problematic. The correlation between the snowfall and its simulated snow depth and soil temperature should be somewhat consistent. As you indicated in section 4, the effects of inputs are limited.

P12, L21-24: "the average … of Fig. 4." The authors seem like to provide a single criterion (one ratio) to assess the behavior of the models. Observations show clearly the difference between deep and shallow snow conditions. It would be better to assess the models for both deep and snow conditions, and Fig. 4 already show such results.  In this paragraph, the "stronger relationship" means "higher correlation

coefficient" or "larger slope in the regression equations"? The term "gradient" used in the abstract and here actually means the slope of the regression between $T_{soil}$ and $T_{air}$. Gradient between $T_{soil}$ and $T_{air}$ can be misunderstood as changes of temperature from soil to air. Probably it is better to indicate its true meaning (slope of the regression, or the ratio between $T_{soil}$ and $T_{air}$ in winter). It is very similar to the freezing season n-factor used in permafrost modeling. You may compare to the winter n-factors used by others.

**Minor points**

P.3, L2: revise "modelling" to "modeling"

P.3, L6: replace "as expressed by" to "in the", delete the two commas around "(ΔT)".

P.4, L14: references are needed at the end of "… soil temperature" to support the statement.

P.5, L24-25: "these simulated relationships": it is not clear what do you mean about "these relationships" without read the entire paper.

P.6, L10: "divided in 14 layers", revise "in" to "into"

P8, L.31-33: "the sentences "We illustrate … 3 regimes." seems can be simplified as "We illustrate the dependence of $T_{soil}$ on $T_{air}$ for three $T_{air}$ ranges".

You used "Larger snow depth", "higher snow depth". Probably can be revised as "thicker snow", or "when the snow is deep", or "with increase in snow depth" etc.

P9, L10: You do not need to redefine the symbols of ΔT and $d_{snow}$ here. Actually, I feel you can replace the word descriptions by the symbols in many places, at least do not need to mention both the word description and symbols.

P9, L29, L31: $\Delta T / d_{snow}$ do mean a ratio as shown in Table 2. Revise "$\Delta T / d_{snow}$ relationship", to "$\Delta T - d_{snow}$ relationship" here and many other places.

P9, L31: "Figure 2 views the $\Delta T / d_{snow}$ relationship in the complementary form of the PDFS of …", revised as "Figure 2 shows the $\Delta T - d_{snow}$ relationship in a complementary form using the PDFS of ……"

P10, L6: "the better models", revise to "the five successful models"

P10, L11: "that affect the air soil temperature difference", revise to "that affect the thermal conductivity of the snow".

P11, L25: "reasonable pattern correlation coefficient with observations", probably means "reasonable spatial pattern of correlation coefficient comparing to that of the observations". L34: "a reverse pattern correlation than observations" revise to "a reverse spatial pattern comparing to that of the observations"

P13, L6: "emphasizing the weakening role of snow depth for $T_{soil}$ under thick snow conditions". Probably should be "emphasizing the reduced sensitivity of $T_{soil}$ to snow depth under thick snow conditions"

Figures: revise "AirT" to "$T_{air}$"

---

## Referee Comment (RC2) · Anonymous Referee #2 · 18 Apr 2016

[referee-annotated manuscript omitted]

---

## Author Comment (AC1) · 20 Jun 2016

Dear Reviewer,

Thanks for the reivewer's comment which we think have helped a lot with the improvement of our paper !  The reply to the reviewers' comment, the track-version of text, revised figures and supplementary are all included in the "zip" file. We kindly ask the reviewer to download the ".zip" file to see specific replies and changes.

Thanks again for your effort ! Best Regards, Wenli

Please also note the supplement to this comment:

[Figure]

http://www.the-cryosphere-discuss.net/tc-2016-36/tc-2016-36-AC1-supplement.zip
* * *

---

## Author Comment (AC2) · 20 Jun 2016

Dear Reviewer,

Thanks very much for your comment, which we agree have helped a lot with the improvement of this paper ! The reply to the reviewers' comments, track version of text, revised figures and summplemantary are all included in the ".zip" file. We kindly ask the reviewer to download the ".zip" file to see the specific replies and changes.

Thanks again for your efforts ! Best Regards, Wenli

Please also note the supplement to this comment:

[Figure]

http://www.the-cryosphere-discuss.net/tc-2016-36/tc-2016-36-AC2-supplement.zip
* * *

---

## Author Response (AR1)

**Response to comments of reviewer #1:**

**General comments**

**Snow and its insulation effects are critical for accurately simulating soil temperature and permafrost in high latitudes. This paper assessed the skills of nine land surface models based on the response patterns of $T_{Soil}$ and the difference of $T_{Soil}-T_{air}$ to snow depth in winter in high latitudes. The observed patterns at 268 climate stations in Russia were used as ground truth. Such an assessment is better than direct point-by-point comparison with station observations. It reveals some structural issues of the models in simulating snow depths and its insulation effects on soil temperature. The results from the observation stations are interesting as well. The data source is solid, the results and analysis are detailed and well presented in most parts. It is worthy to be published.**

We thank Reviewer 1 for the positive feedback on our paper and the useful specific comments that helped improving our manuscript. Please find **the reviewers comments in bold**, our point-by-point answers without formatting, and *changes to the initial manuscript in Italics*.

**Specific comments**

**1. The authors put several lines in abstract about near-surface permafrost. However permafrost results were not described in results and discussion sections, and it is only mentioned in summary and conclusion section. A somewhat proportional amount of description (in terms of length or importance) should be given in results and discussion sections so that it can be included in the summary and abstract. You need to add at least one paragraph about permafrost in the result (as suggested below) or in discussion sections.**

Done. We agree and accordingly we add an additional section "*5. Permafrost area*" in the results section in the paper which include a new table *Table 4*. This new table shows the simulated permafrost area for the nine models.

The new section "5. Permafrost area" reads: "*Snow cover plays an important role in modulating the variations of soil thermodynamics, and hence near-surface permafrost extent (e.g., Park et al., 2015). Here we evaluate if there is a simple relationship between the simulated Northern hemisphere permafrost area and the sophistication and ability of the snow insulation component in the LSM to match observed snow packs. The simulated near-surface permafrost area varies greatly across the nine models in the hindcast simulation (1960-2009; Table 4). Some of the better performing snow insulation effect models (CLM4.5, JULES) simulate a near-surface permafrost area of 13.19 to 15.77 million $km^2$, which is comparable with the IPA map estimate (16.2 million $km^2$) (Brown et al., 1997; Slater and Lawrence, 2013). CoLM and ORCHIDEE, identified as reasonable models with respect to snow insulation, simulate much lower (7.62 million $km^2$) and higher (20.01 million $km^2$) areas, respectively. The main deficiency of CoLM is its too small soil depth (3.4 m) compared with CLM4.5 (45.1 m) despite having very similar snow modules (Table 1). However, ISBA, one of the two models that showed rather limited skill in representing snow insulation effects, also simulates the highest permafrost area (20.86 million $km^2$). This is inconsistent with previous studies (e.g., Vavrus, 2007; Koven et al., 2013) which concluded that the first-order control on modelled near-surface permafrost distribution is the representation of the air-to-surface soil temperature difference. Table 4 shows that the situation is more complex and that snow insulation simulation is not the dominant factor in a good permafrost extent simulation. When the land surface models having poor snow models are eliminated, the remaining models' simulated*

*permafrost area show little or no relationship with the performance of the snow insulation component, because several other factors such as differences in the treatment of soil organic matter, soil hydrology, surface energy calculations, model soil depth, and vegetation also provide important controls on simulated permafrost distribution (e.g., Marchenko and Etzelmüller, 2013)."*

Accordingly, we shortened the permafrost part in the "6. Summary and conclusions" section. The according paragraph reads: *"Snow and its insulation effects are critical for accurately simulating soil temperature and permafrost in high latitudes. The simulated near-surface permafrost area varies greatly across the nine models (from 7.62 to 20.86 million $km^2$). However, it is hard to find a clear relationship between the performance of the snow insulation in the models and the simulated area of permafrost, because several other factors e.g. related to soil depth and properties and vegetation cover also provide important controls on simulated permafrost distribution."*

**2. P.9: Before analyzing the $T_{air}$-$d_{snow}$-$T_{soil}$ relationship, it would be interesting to briefly describe the modeled distribution and errors in snow depth and soil temperature comparing with observations in Russia. The section 4.1.2 about snow depth can be moved to here (table 3 in supplementary can be moved to here as well), and add something similar about the soil temperature. With the soil temperature results, you may add the results of permafrost extent and distribution as you mentioned in the summary and abstract. You may well aware and it is worthwhile to emphasize that the simulated snow depth and soil temperature could be influenced by inputs of the model, and the station observations have limitations in spatial coverage (covers only part of Russia, and may not well represent the grids). However, the response patterns of $T_{soil}$ and $T_{soil}$- $T_{air}$ to $d_{snow}$ should be consistent and can reveal deeper structural issues of the models.**

We appreciate the effort and thought of the referee to suggest some reorganization of the paper structure. However, we do want to keep it as is principally because the main scope of the paper is to evaluate the relationship between air and soil temperatures and its modulation by snow depth and climate regimes. Therefore, we present in the "Results" (section 3) the results for the relationships. This is then followed by the in-depth discussion of the indicated across-model differences in the relationships and its influencing factors (section 4). Here, snow depth comes into play and is discussed in detail (section 4.1.2). There we also discuss the relationship between the input precipitation/snowfall and simulated snow depth. We agree that snow depth is important for the snow insulation effect; therefore we followed your suggestion and show now for the snow depth both the spatial patterns (Fig. 6) and the station-based bias statistics. This means we moved earlier SI Tab. 3 into the new *Table 3*. In accordance with this, we moved also the ΔT bias statistics into the text (means we moved the whole earlier SI Tab. 3 to the new *Tab. 3*), because ΔT is also in detail discussed in section 4. We also followed your suggestion to present the permafrost extent (new *section 5* and new *Tab. 4*); see our above answer to your related comment #1). We do present information about soil temperature biases. Actually, the presented bias statistics for both air temperature (SI Tab.2) and for ΔT (*Tab.3*) give the information about soil temperature bias. An additional table in the manuscript would be redundant information. For your convenience, we add here explicitly the $T_{soil}$ bias table:

| $T_{soil}$ bias statistics (n=479); $StDev^{Obs}$=1.5 K | | | |
|---|---|---|---|
| | BIAS (K) | StDev (K) | RMSE (K) |
| CLM45 | -2,4 | 1,2 | 3,5 |
| CoLM | 1,8 | 1,0 | 2,9 |
| ISBA | -9,4 | 1,8 | 9,9 |

| | | | |
|---|---|---|---|
| JULES | -3,1 | 2,6 | 4,5 |
| LPJ-GUESS | -1,0 | 0,9 | 3,2 |
| Miroc-ESM | -2,5 | 2,1 | 4,8 |
| ORCHIDEE | -6,1 | 1,8 | 6,6 |
| UVic | -6,2 | 1,8 | 7,0 |
| UW-VIC | -1,8 | 1,4 | 4,0 |

Further, Figs. 4 and 5 present also information of the simulated $T_{soil}$ in comparison to station observations. For example, both figures clearly show the strongest underestimation of $T_{soil}$ in ISBA, as shown in the table above too. Finally, we emphasize that the focus of the paper is on the relationships or functional behaviors. Even if $T_{soil}$ is biased, the relationship between $T_{soil}$ and $T_{air}$ can be well represented compared with observations. For example, CLM45 and JULES have a cold bias in $T_{soil}$ (too cold by ca. 3 K), but can represent the dependency of the $T_{soil}$-$T_{air}$ relationship on snow depth regime well (Fig.4; Tab. 2).

Yes, the station data set covers the Russian Arctic. We focus on this because this data set was compiled within PCN project which initiated this model intercomparison study. However it is important to emphasize that the spatial coverage of the 579 stations reporting snow depth, 268 stations reporting simultaneously air and soil temperatures and snow depth, and 518 stations reporting air temperature is quite good (see Figs. 3, 6, SI figures) to cover the model grid boxes of 0.5x0.5 deg. And indeed, the presented response patterns allow a much better assessment than direct point-by-point comparison with station observations. Therefore we have chosen this approach. We follow your suggestion and highlight this better by adding a sentence in the "Summary and conclusions" section: "*The presented relation diagrams of $T_{soil}$ and the difference of $T_{soil}$-$T_{air}$ to snow depth allow a much better assessment to reveal structural issues of the models than a direct point-by-point comparison with station observations.*"

**3. P.8, L. 11-13: "We assume that …in winter". I feel such an assumption is not necessary. The effects of soil moisture and texture do have effects but is much smaller than that of snow. You may revise it to "The effects of other factors on ΔT are much smaller than that of snow" or delete the sentence.**
We agree and have revised this to: "*In winter, the effects of other factors (e.g. soil moisture, texture) on ΔT are much smaller than that of snow.*"

**4. P. 12, L. 2-5: This sentence does not connect well with the previous one (why LPJ-GUESS produces very low correlation coefficients). In addition, the meaning of the sentence is problematic. The correlation between the snowfall and its simulated snow depth and soil temperature should be somewhat consistent. As you indicated in section 4, the effects of inputs are limited.**
We agree and deleted this sentence.

**P. 12, L. 21-24: "the average … of Fig. 4." 1) The authors seem like to provide a single criterion (one ratio) to assess the behavior of the models. Observations show clearly the difference between deep and shallow snow conditions. It would be better to assess the models for both deep and snow conditions, and Fig. 4 already show such results. 2) In this paragraph, the "stronger relationship" means "higher correlation coefficient" or "larger slope in the regression equations"? The term "gradient" used in the abstract and here actually means the slope of the regression between $T_{soil}$ and $T_{air}$. Gradient between $T_{soil}$ and $T_{air}$ can be misunderstood as changes of temperature from soil to air. Probably it is better to indicate its true meaning (slope of the regression, or the ratio between $T_{soil}$ and**

**Tair in winter). 3) It is very similar to the freezing season n-factor used in permafrost modeling. You may compare to the winter n-factors used by others.**

1) It seems to us that there is probably a misunderstanding. Indeed, this whole second paragraph in section 3.2 evaluates the behavior of the model's $T_{air}$ vs. $T_{soil}$ relation under both thick and thin snow conditions. And, we assess the simulated relationships by different measures. First, we compare the simulated slopes of the $T_{air}$ vs. $T_{soi}$ relationship under thin and thick snow with the according slopes from the observed relationships. This quantification is given in Table 2. Second, we calculate the RMSE between the observed and modeled relationship. These numbers are given in each model panel in Fig. 4 for both thin and thick snow. And third, we calculate the ratio of the slopes under these two snow regimes (ratio of slope under shallow snow divided by that of thick snow). All three criteria give a solid evaluation of the models relationships and they quantitatively confirm each other by arriving at the same conclusion: some specific model's behavior under thick and thin snow is in agreement, other models cannot reproduce the observation.

It seems we were not enough clear and improved this paragraph accordingly. It reads now:

*Figure 4 clearly shows that some models (CoLM, CLM45, JULES) can capture this modification of the Tair-Tsoil relation by snow depth regime well. Their regression slopes for thick and thin snow are well separated and in agreement with those from the observed relationship (Table 2). The RMSE of their modeled Tsoil vs. Tair relationships from observations is smaller than 4 °C. These models better reproduce the observed ΔT vs. dsnow relationship. Other models (LPJ-GUESS, MIROC-ESM, ORCHIDEE) strongly underestimate the increase of the Tsoil vs. Tair regression slope for decreasing snow depth. They also produce a regression slope for thick snow more than twice as large as observations. Two models (ISBA, UVic) fail here and do not show any sensitivity in the Tair-Tsoil relation to snow conditions (Fig.4, Tab.2). Another measure quantitatively confirms the same models behavior: The observed average dsnow in the shallow snow regime is 13.7 cm and that for the thick snow regime is 58.5 cm, so we would expect, if near-surface air temperature and conductivities were equal in both snow depth classes, a ratio between the slopes for shallow and thick snow of 4.3. CLM4.5, CoLM, and JULES reproduce this observed variation in the Tsoil vs Tair relation better than others (Table2). JULES and CoLM indicate a factor of 4 change, while CLM4.5 indicates a factor of 2 change. Other models (LPJ-GUESS, MIROC-ESM, ORCHIDEE) strongly underestimate the increase of the regression slope for decreasing snow depth; they simulate only a factor change of about 1.5. The two models that had also unrealistic ΔT vs dsnow relationships (ISBA, UVic) also fail in this evaluation of their Tsoil vs Tair relationship. They simulate a too strong sensitivity of Tsoil to Tair (regression slopes larger than 0.9 °C/ °C, $R^2$>0.7; Table 2) that are almost completely independent of the snow depth regimes, particularly in ISBA, which is not consistent with observations. These models' spatial correlation patterns between Tsoil and Tair also differ greatly from the observations and the other models (SI Fig. 3) and show very high positive correlation (r > 0.8) in most regions, as may be expected from the large regression slope shown in Fig. 4. The RMSE of their modeled Tsoil vs Tair relationships from observations reaches ca. 10 °C.*

2) We agree and we changed the wording through all the manuscript. We use either "larger slope in the regression between $T_{soil}$ and $T_{air}$" or "larger regression slope", or "stronger sensitivity of $T_{soil}$ to $T_{air}$".

3) Yes, we agree and included one paragraph "*This is consistent with observations that the mean freezing n-factor (the ratio of freezing degree days at the ground surface to air freezing degree days) is high at sites where the snow cover is thin or absent, and low at sites where the snow cover is thick (e.g., for Yukon Territory in Canada; Karunaratne and Burn, 2003).*"

**Minor points**

**P.3, L.2: revise "modelling" to "modeling"**
Done.

**P.3, L.6: replace "as expressed by" to "in the", delete the two commas around "($\Delta T$)".**
Done.

**P.4, L.14: references are needed at the end of "… soil temperature" to support the treatment.**
Done. This has been clarified; this sentence is related to the sentence and references before.

**P.5, L. 24-25: "these simulated relationships": it is not clear what do you mean about "these relationships" without read the entire paper.**
Done. We revised it to: "*What is the range of the simulated air-soil temperature relationship across the model ensemble?*"

**P.6, L.10: "divided in 14 layers", revise "in" to "into"**
Done.

**P.8, L.31-33: "the sentences "We illustrate … 3 regimes." seems can be simplified as "We illustrate the dependence of $T_{soil}$ on $T_{air}$ for three $T_{air}$ ranges".**
Done. It reads now "*We illustrate the dependence of $\Delta T$ and $T_{soil}$ on $d_{snow}$ for three $T_{air}$ ranges.*"

**You used "Larger snow depth", "higher snow depth". Probably can be revised as "thicker snow", or "when the snow is deep", or "with increase in snow depth" etc.**
Done.

**P.9, L.10: You do not need to redefine the symbols of $\Delta T$ and $d_{snow}$ here. Actually, I feel you can replace the word descriptions by the symbols in many places, at least do not need to mention both the word description and symbols.**
We would like to keep it here at the beginning of the "Results" section, just for reminder the reader. We also keep it in first paragraph of "Summary and conclusions". But, except this, we followed your suggestion and replaced the word descriptions by the symbols in the manuscript.

**P.9, L.29, L.31: $\Delta T/d_{snow}$ do mean a ratio as shown in Table 2. Revise "$\Delta T/d_{snow}$ relationship", to "$\Delta T$-$d_{snow}$ relationship" here and many other places.**
Done. We do agree. We replaced "/" by "vs." at all the respective places in the whole manuscript.

**P.9, L.31: "Figure 2 views the $\Delta T/d_{snow}$ relationship in the complementary form of the PDFS of …", revised as "Figure 2 shows the $\Delta T$-$d_{snow}$ relationship in a complementary form using the PDFS of ……"**
Done. We also change "$\Delta T$-$d_{snow}$" to "$\Delta T$ vs. $d_{snow}$". See above answer.

**P.10, L.6: "the better models", revise to "the five successful models"**
Done.

**P.10, L.11: "that affect the air soil temperature difference", revise to "that affect the thermal conductivity of the snow".**
Done.

**P.11, L.25: "reasonable pattern correlation coefficient with observations", probably means "reasonable spatial pattern of correlation coefficient comparing to that of the observations".**
Done. Actually, here we refer to the similarity between the simulated spatial patterns and the spatial pattern from observations. For this, we calculated the spatial pattern correlation coefficient. To be more precise, we improved this sentence to "*… show a reasonable spatial pattern correlation coefficient …*".

**P.11, L.34: "a reverse pattern correlation than observations" revise to "a reverse spatial pattern comparing to that of the observations"**
Done.

**P.12, L.6: "emphasizing the weakening role of snow depth for $T_{soil}$ under thick snow conditions". Probably should be "emphasizing the reduced sensitivity of $T_{soil}$ to snow depth under thick snow conditions"**
Done.

**Figures: revise "AirT" to "$T_{air}$"**
Done.

**Response to comments of reviewer #2:**

We thank Reviewer 2 for providing valuable comments on our paper that helped improving our manuscript. Please find **the reviewers comments in bold**, our point-by-point answers without formatting, and *changes to the initial manuscript in Italics*.

**Comments:**

**1. P5 L22-24: Most of the analysis is based on using the meteostation data across the Russian territory from 1980-1990. Unfortunately, there is no in-depth analysis of the observations and why these data can be used. Observations at many meteostations are performed at the disturbed conditions and thus only a small subset of this data could be used in the direct comparison. The authors simply assume that just all data could be used. The meaningful comparison of the model to observations at the disturbed sites could be done if the model also simulates the disturbed conditions. One of the major problems in this paper is that many observations were taken at the disturbed sites near meteostations, while models simulate typical ground conditions. To do the comparison correctly it is either necessary to model disturbed conditions in the models or remove 'corrupted' observations from the analysis.**

This data set is quality checked and officially released by the All-Russian Research Institute of Hydrometeorological Information-World Data Centre (RIHMI-WDC; http://meteo.ru/). They provide quality controlled soil temperature at depths to 320 cm from meteorological stations of the Russian Federation. The data was subject to quality control by using four methods of statistical control (histograms, standard deviation check, check for connectedness of values that are neighboring in time).

Meteorological sites are located in an open and typical place within the surrounding terrain. They are located far from the major obstacles and water bodies that may have a direct effect on the quality of measurements. To keep the surface of the meteorological site in its natural state, it is permitted to walk only on the specially laid tracks, a width not exceeding 40 cm, within the site territory. In the warm season grass on the site is regularly mowed and trimmed. The height of the grass on the site is not more than 20 cm. In the winter, the natural state of the snow cover is not broken. Observing conditions at the Russian stations in all meteorological elements correspond with WMO standards. The observations presented have been included in data sets, such as GSOD, HadSRUT4 etc. and are widely used in climate research.

Soil temperature measurements are carried out simultaneously with the measurements of the whole complex of meteorological observations (temperature, characteristics and dynamics of snow cover and precipitation and so on). The soil temperature observations are under the original surface (from draw-out thermometer data). All meteorological observations, including soil temperature, are produced exclusively within the same site area (26 by 26 meters in size), and under constant careful control of workers who are caring for a site. This prevents any accidental violations of the integrity of the site and guarantees the quality of observations. A detailed history of the development of methodology for soil temperature measurements was provided by Bykhovets et al. (2007). A detailed description of dataset preparation is provided in Sherstiukov (2007). The archived soil temperature dataset was run through four independent methods of quality control (Sherstiukov, 2012).

Thus, the data of meteorological observations at the Russian stations for soil temperature can be effectively used in the evaluation of thermal changes in the upper layers of permafrost zone, as well as in the analysis of processes of thermal interaction between the atmosphere and soil. Similar conclusions were obtained by leading Russian scientists in this area (Anisimov and Sherstiukov, 2016; Pavlov and Malkova, 2009) and internationally (Park et al., 2014; Brun et al., 2013; Decharme et al. 2016; PaiMazumder et al., 2008).

However, as Park et al. (2014) pointed out, the observations at some locations could have been unavoidably disturbed by grass cutting during the warm season and the removal of organic materials, mainly at agricultural sites. These disturbances may cause increased warming of the soil over time. Therefore, long-term soil temperature trends in could potentially include this non-climatic component (Frauenfeld et al 2004).

Thus, we agree that we have to be careful which and how we use the observation data. But we can argue that our study results are solid. First, and most important is that we do not present either direct point-by-point comparison with station observations nor trends, but we investigate variables relationships. We evaluate the models functional behaviors (inter-variable scatter plots, Figs. 1, 4, 5 and PDFs, Fig. 2). The inter-variable scatter plots (which show medians and the 25th and 75th percentiles) as well as the PDFs present the overall functional behavior of snow insulation effects, and not the individual station's representation. Therefore, this kind of benchmark to evaluate the models skill is a solid approach using the station data, which include uncertainties (by showing the percentiles). This approach, as the other reviewer points out, reveals some structural issues of the models in simulating snow depths and its insulation effects on soil temperature. Secondly as we are only looking at upper soil temperatures, any change due to changes in soil column temperature gradient resulting from grass removal are negligible. Thirdly, the agriculture sites concerns are not an issue for our permafrost area focused study.

According to the reviewer's comment, we include in section 2.2 two references for more details of the data set: "*A detailed description of dataset preparation is provided in Sherstiukov (2012a). Observing conditions at the Russian stations in all meteorological elements correspond with WMO standards. The observations presented have been included in data sets, such as GSOD, HadSRUT4 etc. and are widely used in climate research (e.g. Anisimov and Sherstiukov, 2016; Decharme et al. 2016; Park et al., 2014; Brun et al., 2013; Pavlov and Malkova, 2009; PaiMazumder et al., 2008).The soil temperature dataset was run through four independent methods of quality control (Sherstiukov, 2012b).*"

Further, we emphazise the point that possible disturbances do not affect our results:
"*However, some soil temperature observations could be disturbed by grass cutting during the warm season and the removal of organic materials, mainly at agricultural*

*site, which may affect the trend in warm season (Park et al., 2014), but this does not affect our results about the air- upper soil temperature relationship in winter.*"

Finally, we cannot compare the model results and observation over only bare ground (or short grass) in the PCN simulations. Only the grid-cell average results were kept (and not the value for each vegetation and bare ground tile).

To show you that this parameter choice does not affect our results, Fig. X1 is the comparison with snow depth from GlobSnow using a density of 300 kg m⁻³. This figure confirms that the pattern does not change and shows that the differences in snow depth are small (less than 10 cm).

[Figure]

Figure X1: Snow depth (cm) derived from GlobSnow SWE using different assumptions of snow density, winter 1980-2000.

Further, our calculations show that the model biases in snow depth are quite similar using either the station or the GlobSnow data (see earlier SI Tab.3, now moved to Table 3). Also, if we re-calculate the model biases with respect to GlobSnow derived with a density of 300 kg m-3, the biases change only slightly and the model ranking is unaltered. For your convenience, we list here the comparison of the RMSE for snow depth (cm):

| | RMSE of snow depth (cm) | | |
|---|---|---|---|
| | GlobSnow | | Stations |
| | Density=250 kg/m$^3$ | Density=300 kg/m$^3$ | |
| CLM4.5 | 18.1 | 20.5 | 18.1 |
| CoLM | 22.1 | 24.9 | 21.4 |
| ISBA | 19.8 | 22.9 | 18.8 |
| JULES | 12.8 | 11.5 | 14.1 |
| LPJ-GUESS | 16.0 | 13.7 | 17.3 |
| MIROC-ESM | 14.0 | 13.7 | 17.9 |
| ORCHIDEE | 15.3 | 12.4 | 16.5 |
| UVic | 16.8 | 16.5 | 18.9 |
| UW-VIC | 20.0 | 22. 6 | 19.8 |

**4. P7 L29: "the GlobSnow product can show regional differences (of ca. 0.5-5 cm) with biases increasing with increasing SWE" Is it a systematic bias in conversion of the SWE to snow height? If so, why not to take it into the account. Please specify where the regional differences are the largest, what areas have smallest. How many stations are located in the areas where differences are largest. Also, please list differences in % not the absolute values, since in some locations 5cm could be 25% of error vs 10% of error.**

We do not aim to evaluate the GlobSnow data. There are quite a lot of papers about details of GlobSnow validation and uncertainties. The given sentences are from the references. The paper is already rather long and we do not feel a digression into this data set is of interest to most readers. Again we emphasize that all the relationship analysis we do is based on the station data (Figs. 1-5). We use GlobSnow only for the evaluation of the simulated spatial maps of snow depth (Fig.6). This has been done to support the station data results, and to arrive at a more solid comparison with the models. And indeed, we show that the calculated model biases in snow depth are quite similar using either the station or the GlobSnow data (see earlier SI Tab.3; now *Table 3*). To make this clearer in the text, we now directly cite the sentence from the references and also include a reference related to our considered Russian Arctic or Eurasia region. We improved the according part in section 2.2: *"...GlobSnow-2 product (http://www.globsnow.info/swe/), which has been produced using a combination of passive microwave radiometer and ground-based weather station data (Takala et al., 2011). Orographic complexity, vegetation cover, and snow state (e.g. wet snow) affect the accuracy of this product. When compared with ground measurements in Eurasia, the GlobSnow product can show root-mean-square error (RMSE) values of 30 to 40 mm for SWE values below 150 mm, with retrieval uncertainty increases when SWE is above this threshold (e.g., Takala et al., 2011; Muskett, 2012; Khelemet et al., 2013)."*

**5. P8 L11-13: "We assume that there is relatively little impact due to soil moisture and texture between surface and 20 cm depth in winter." This is probably true for the cold climate conditions, while in the warm climate the surface might freeze, but 0.20m could stay thawed.**

Yes, we consider here cold climate winter conditions. We follow the other reviewer's suggestion to formulate it "*In winter, the effects of other factors (e.g. soil moisture, texture) on $\Delta T$ are much smaller than that of snow.*"

**6. P8 L13-14: "Although we recognize the difference between ground surface and 20 cm soil temperatures and that soil organic layer could play a role in certain locations". What certain locations? In warm climate? Please be specific.**

We do agree that the snow effect is the key in winter cold climate. You (and the other reviewer) also emphasize, other factors like soil moisture, texture, organic matter are of secondary importance, and we agree on this. Therefore, we deleted this sentence to avoid confusion.

**7. P8 L17: "However, we find that the results do not significantly change when the model simulated temperature differences between ground surface temperature (GST) and near-surface air temperature (Tair) are used instead of between 20 cm soil (T20cm) and near-surface air temperatures." Again please be specific. How much is not significant?**

This sentence is to confirm that our results of the functional behavior (Figs. 1-4) do not qualitatively change if we use either the $\Delta T$=Tsoil-Tair or $\Delta T$=GST-Tair. It is to tell the reader that we checked this. To make it clear, we do not have GST observations! Therefore, we have to use Tsoil. The question is how sensitive the results would be when using GST instead of upper soil-layer temperature (Tsoil at 20 cm depth). This would give an indication if/how soil characteristics are important for our presented relationships. The only thing we can do for this is to look into the model results. For your convenience, we show here one example plot (Fig. X2). The comparison of the red ($\Delta T$=Tsoil-Tair) and green ($\Delta T$=GST-Tair) clearly show that the regression does not materially change. "*To test how sensitive are results using 20 cm temperatures instead of ground surface, we also analysed model simulated temperature differences between ground surface and Tair, and found not qualitative differences, hence justifying use of 20 cm observations.*"

[Figure]

Figure X2: Variation of ΔT (K) with snow depth (cm) under cold conditions (Tair ≤ -25 ℃) for winter 1980-2000. The dots represent the medians of 5 cm snow depth bins, calculated from all Russian station grid points (n=268) and 21 individual winters. Red: ΔT is the difference between soil temperature at 20 cm depth and Tair, winter (DJF); Green: ΔT is the difference between ground surface temperature and Tair, winter (DJF); Orange: ΔT is the difference between soil temperature at 20 cm depth and Tair, winter (NDJFMA).

**8. P8 L4: "we checked that a different winter definition (NDJFMA) does not substantially change the results" What is substantial, please be specific.**
This sentence is to confirm that our results of the functional behavior (Figs. 1-4) do not change if we use another definition for the winter season. It is to tell the reader that we checked this. The question is how sensitive the results would be when using a different winter definition, because we know that snow can begin in November and end at the beginning of May. For your convenience, we show here one example plot (Fig. X2). The comparison of the red (winter-DJF) and orange (winter-NDJFM) clearly show that the regression does not materially change. We deleted "substantially" and substitute "qualitatively". "*Our analysis is focused on the common winter (DJF) condition, although snow can begin in November and end at the beginning of May, but we checked that a different winter definition (NDJFMA) does not qualitative change any of the inter-variables relationships found.*"

**P3 L5-6: delete "as expressed by simulated differences"**
Done.

**P4 L5: Please try to avoid parentheses, usually it is possible to write the manuscript without them.**
Done.

**P4 L7-8: delete "quality" and parentheses.**
Done.

**P4 L14, "soil temperature", reference is missing.**
The reference is provided just in the former sentence.

**P4 L22-23: Please rewrite without parentheses.**
Done.

**P8 L21: "We use correlation analysis to investigate the co-variability…". Please provide formulae or references to this analysis**
Done. "*We use the Pearson product-moment correlation coefficient and its significance (von Storch and Zwiers, 1999)…* "

**P8 L22: "The input consists of detrended time series of winter means at each grid point." How did you compute detrended time series? You may add details to the appendix.**
Done. "*Before we compute the correlations we detrended the data by removing a least squares regression line.*"

**P8 L26: "Student t-test" Reference?**
Done. "*… coefficients is estimated by the Student's t-test (von Storch and Zwiers, 1999).*"

**P8 L28: "To further examine the functional behavior between different variables, we present relationship…". What variables? Please be specific.**
Done. We introduce the key variables in the paragraph above. "*… 4 key variables: near-surface air temperature ($T_{air}$), near-surface soil temperature (soil temperature at 20 cm depth; $T_{soil}$), snow depth ($d_{snow}$), and the difference between $T_{soil}$ and $T_{air}$.*"

**P8 L31: "We illustrate the dependence on air temperatures by evaluating".**
**Dependence of what? What variable? Please be specific**

Done. *"We illustrate the dependence of ΔT vs. dsnow and $T_{soil}$ vs. $d_{snow}$ relations for three Tair ranges."*

**P9 L2-3: "The principal motivation for such classifications is to distinguish dry snow pack regimes from those where sporadic melt may occur even in winter." Usually, the motivation or idea to do something goes first. The next sentence describes how this motivation is implemented. Please re-write this paragraph such that the motivation is at the beginning.**

Done. We revised this according to your suggestion. We start with *"To distinguish dry snow pack regimes from those where sporadic melt may occur even in winter, we split Tair into 3 regimes: the coldest conditions… "*

**P9 L11: "observations and all models produce a clear relationship" I do not see this "clear relationship" at the first glance in Figure 1. Please be specific. Describe what the reader may find on this figure and then state the relationship.**

Done. Fig. 1 clearly indicates that the observations as well as all models show an increase of ΔT with increasing dsnow. We improved this paragraph. First we start with the observations *"The air-soil temperature difference (ΔT) - snow depth ($d_{snow}$) relationship in winter (Fig. 1) shows in the Russian station observations an increase of ΔT with increasing $d_{snow}$. The data exhibit a linear relation between ΔT and $d_{snow}$ at relatively shallow snow depths with a trend towards asymptotic behavior at thicker snow, which is in agreement with earlier findings (Zhang, 2005; Ge and Gong, 2010; Morse et al., 2011)…".* Then we continue with the models behavior *"All models reproduce the observed relationship, i.e. increasing ΔT with increasing $d_{snow}$. However, Fig. 1 also shows a wide across-model spread in the simulated relationships, and that some of the models are not consistent with the behavior in the observations…"*

**P9 L14: "that some of the models are not consistent with the behavior in the observations. There is also significant scatter in the observation-based relationship, the inter-quartile range…". Observations are just plotted on the last panel in figure 1. It takes time to find the observations on figure 1. Please try to re-design Figure 1 and all other figures such that observations stand out and could be easily noted. And please point out/circle this scatter on the figure.**

Sorry, we do not agree that it is hard to recognize the observations in the figures, though obviously this a personal issue. However, we consistently show observations in the bottom subpanel of our figures. To highlight the scatter in addition to the median (which is presented by the dots) we plot therefore upper and lower bars on these dots, indicating the 25th and 75th percentiles.

**P9 L21-24: "The Russian station data and some model results exhibit a linear relation...larger snow depths (Fig. 1), which is in agreement with earlier findings (Zhang, 2005; Ge and Gong, 2010; Morse et al., 2011)." Move this sentence up to**

**the beginning of the above paragraph.**

Done. We largely revised the discussion in Section 3.1, such that we discuss observations first, and then followed by the model representation.

**P9 L31-32: "Figure 2 views the…relationship in the complementary form of the PDFs …different snow depth and air temperature regimes". Before showing the PDF, please explain what you plan to analyze and how the PDF can help you to achieve this goal. It is really confusing for a reader to understand why PDF are now considered and plotted.**

Done. We revised the introduction part of this analysis: "*Figure 2 views the $\Delta T$ vs. $d_{snow}$ relationship in a complementary form using the PDFs of $\Delta T$ for different snow depth regimes. This analysis allows a detailed evaluation of the snow regime-dependent $\Delta T$ separation by quantifying and comparing the modal value and width of the different conditional PDFs.*"

**P10 L1: "we divide the data into "shallow" ($d_{snow} \leq 20cm$) and "thick" ($d_{snow} \geq 45cm$) regimes". Why do you choose these thresholds to characterize shallow and deep snow?**

Done. We explain that the Russian snow depth data show a non-normal distribution with a median value of 30 cm (SI Fig. 1). Therefore, we have chosen two classes, one with a threshold below ($d_{snow} \leq 20cm$) and one with a threshold above ($d_{snow} \geq 45cm$) of this median value. "*Since the Russian snow depths are clearly non-Normal in distribution (SI Fig. 1, with a median $d_{snow}$ of 30 cm), we divide the data into "shallow" ($d_{snow} \leq 20$ cm) and "thick" ($d_{snow} \geq 45cm$) regimes to separate two snow depth regimes.*"

**P10 L3: "Based on the $\Delta T$ PDFs". There are lots of line of figure 2. Maybe a revision to figure 2 is necessary. Could you please be specific how these five models separate the regimes, while others do not?**

Done. We agree that Fig. 2 was too busy. We improved the readability of Fig.2 by splitting it into two figures *Fig.2a and Fig.2b*. Fig.2a shows the PDFs for snow depth classes, while Fig.2b shows the PDFs for different air temperature regimes. We also use now the same colors for the different snow classes as in Fig. 4 to be consistent. The color of the different Tair classes is consistent in all relevant figures (Figs. 1, 2, 5). We improved the discussion of the separation of the PDFs: "*Based on the $\Delta T$ PDFs, five models (CoLM, CLM4.5, JULES, ORCHIDEE, MIROC-ESM) successfully separate the $\Delta T$ regimes under different snow depth conditions. Their simulated $\Delta T$ PDFs have a smaller modal value for thin snow than for thick snow, as in the observations. The other models clearly fail in separating the $\Delta T$ PDFs for the two different snow depth regimes.*"

**P10 L9-17: This is mainly a description of observations. I would suggest to fold in this paragraph into the text right after the introduction of figure 1.**

Thanks, but we disagree because: First we discuss the general functional behavior of

ΔT vs. dsnow relationship, i.e. increasing ΔT with increasing dsnow (Fig.1). Here we start with observations, followed by model evaluation. This is then further supported and discussed by Fig.2a. Next we discuss then the impact of Tair on the ΔT vs. dsnow relationship (different colored curves in Fig. 1 and Fig.2b). Therefore, we keep this paragraph there at its place, because this is the introduction/motivating paragraph, why we expect and look at the impact of Tair on the the ΔT vs. dsnow relationship. To make this clearer, we add at the end of this paragraph *"Therefore, we can expect that the same thickness of snow in colder climates will provide greater insulation than it would in warmer climates."*

**P10 L19-20: "The observations in Figs. 1 and 2 indicate that snow under colder climates have greater insulation than under warmer climates." Please add an opening sentence to this paragraph. It would be great to give a reader a small hint about what is going to follow in the text below. This seems to be very typical for this manuscript, starting from the details and then reveal a motivation behind all of this.**

Actually, we do have an opening paragraph above these lines, which gives the motivation looking at the impact of Tair on the the ΔT vs. dsnow relationship. There we explain the hypothesis that snow under colder climates have greater insulation than under warmer conditions, based on known facts: *"Both Figs. 1 and 2 further indicate that air-soil temperature differences are related to air temperature conditions. This is expected due to snow pack properties, particularly its density and moisture content, that affect the thermal conductivity of the snow. For example, the density of fresh fallen snow tends to be much lower under cold air temperatures than warm (Anderson, 1976), leading to increased insulation (larger ΔT). Snow densification is also a function of air temperature, for example, depth hoar metamorphosis of the snow pack, which produces more insulation (loosely packed depth-hoar crystals have very low thermal conductivity), is promoted by strong thermal gradients in the snow pack, and is typical of continental climates (e.g., Zhang et al., 1996). Therefore, we can expect that the same thickness of snow in colder climates will provide greater insulation than it would in warmer climates."* This paragraph gives the motivation and hypothesis. Then, with the next paragraph we discuss that indeed our observational analysis confirm this expected impact of Tair on the ΔT vs. dsnow relationship. According to this comments, we improved the connection between the two paragraphs. The first paragraph is followed by *"Indeed, our observational analysis (Figs. 1 and 2) confirm this. This is shown by a larger ΔT for colder $T_{air}$ (for a certain snow depth) and a greater sensitivity of ΔT to changes in $d_{snow}$ (Fig. 1), and by the larger modal value of the ΔT PDF for colder $T_{air}$ than for warmer $T_{air}$ (21 K for $T_{air} \leq -25$ °C and 9 K for $-15$ °C < $T_{air} \leq -5$ °C; Fig. 2b). This is consistent with colder climates having lower density snow packs, and the differences are in line with measurements of snow density variability (Zhong et al., 2013)."*

**P10 L32: "ΔT/ $d_{snow}$". On page 9, Line 25 there was a different notation for this relationship. I would suggest to make uniform notations. The sign "/" means**

division and maybe there is a better choice, e.g. \Delta{T}(d_snow) P12 L16: "$T_{soil}$ /$T_{air}$" Maybe it is possible to find a better notation. The sign / means division. Could you use T_obs(T_air)

Done. We agree with you. We changed all the "/" with "vs." throughout the paper.

**P11 L5-7: "Our analysis (Fig. 1) indicates that some models (CLM4.5, CoLM, JULES) are better able to…". Please move this section of the text close to figure 1, the reader now needs to go back in text. Everything related to figure 1 needs to stay close to figure 1.**

Thanks for the suggestion, but the structure as it stands seems clearer to us. First we discuss the general functional behavior of $\Delta T$ vs. dsnow relationship, i.e. increasing $\Delta T$ with increasing dsnow (Fig.1). Here we start with observations, followed by model evaluation. This is then further supported and discussed by Fig.2a. Next we discuss the impact of Tair on the $\Delta T$ vs. dsnow relationship (different colored curves in Fig. 1 and Fig.2b). Again, we start with observations, followed by model evaluation. Here we discuss in detail that part of Fig.1 which shows the snow-dependence on $\Delta T$ vs. dsnow relation (colored curves in Fig.1). To make this structure even more clear, we add one opening sentence before this model evaluation part starts. "*If we evaluate the models with respect to this observed impact of Tair to the $\Delta T$ vs. $d_{snow}$ relationship, we demonstrate that some models (CLM4.5, CoLM, JULES) are better able to replicate the effect than others (LPJ-GUESS, MIROC-ESM, ORCHIDEE, UW-VIC) (Fig. 1)…*"

**P11 L22: "strong". I also see lots of green and blue. I guess this is an indicator of weak correlation.**

Sorry, we think this is a misunderstanding; we were not clear enough. Here "strong spatial variability" refers to the pronounced spatial variability in the correlation. We see in Fig.3 in some regions redish color (high correlation) or in other regions greenish color (low correlation). We revised this sentence "*The maps of the $\Delta T$ vs $d_{snow}$ correlations in winter (Fig. 3) demonstrate a pronounced spatial variability in the $\Delta T$ vs $d_{snow}$ relationship.*" And, we re-ordered some text such that this sentence is then directly followed by the description of the details of regional variation in the correlation.

**P11 L23: "but indicate that most models agree on the general large-scale pattern." Please describe what you see on the generated maps and only then state your conclusion. The first sentence is to describe what is going to happen in the rest of paragraph. It appears that the conclusions/details are always are beginning of many paragraphs. of course, there is a style to present a material this way, but it is only good for press-conferences when it is not enough time to describe assumptions, hypothesis, etc.**

Done. We improved this paragraph to make it clearer. We start with the most obvious finding that the maps of the $\Delta T$ vs. dsnow correlations show a pronounced spatial variability in this correlation. Then, we describe the details of the observed spatial variation. This is then followed by the model evaluation. Thus, we do have a clear and consistent presentation structure.

**P12 L9: delete "Previous authors" and parentheses.**
Done. We re-arranged this sentence. *"Previous studies have noted that the strength of relationship between $T_{soil}$ and $T_{air}$ is modulated by $d_{snow}$ and the snow insulation effect increases only up to a limiting depth beyond which extra snow makes little difference to soil temperatures (Smith and Riseborough, 2002; Sokratov and Barry, 2002; Zhang, 2005; Lawrence and Slater, 2010)."*

**P13 L18: "SI Fig.3". L24: "SI Fig.4". Bring this figure from the appendix to the text and describe how this figure is obtained. P15 L32: "SI Table 3". There is no reason to move table and figures to SI when they are cited so often. Ideally, it should be no mentioning of figure from the SI.**
We moved SI Table 3 to the main text; it is Table 3 now. Both figures (SI Fig.3, SI Fig.4) are cited only once now and are supplementary information.

**P13 L20-21: "Obvious outliers in the $T_{soil}$ /$T_{air}$ correlation maps (SI Fig. 3) are ISBA and UVic, which strongly overestimate the correlation (r > 0.9)". What does it physically mean?**
Done. We improved this discussion. *"Obvious outliers in the $T_{soil}$ vs $T_{air}$ correlation maps are ISBA and UVic, which strongly overestimate the correlation (r > 0.9) over most of the Arctic. This indicates an underestimated snow insulation effect, and confirms the weak insulation in both models, which has been initially discussed based on the underestimated ΔT (Fig. 1) and weak correlation between ΔT and $d_{snow}$ (Fig. 3)."*

**P14 L10: "the biases range from -0.8 K to -4.7 K (SI Table 2)". Please state that this is for winter months in the text.**
Done. *"The biases of winter mean air temperature ranges…"*

**P14 L25-26: "underestimate" and "overestimate". Does it depend on the values of 250 kg/m3 used to convert the satellite measured SWE to snow depth.**
No, this is not the case. Please see our answer to the related comment #3. Furthermore, we show in Table 3 (earlier SI Tab.3) that the model biases are consistent regardless of which observations are used for the model evaluation. The model biases with respect to the station data are consistent with those with respect to the GlobSnow data; the model ranking is not affected.

**P14 L28-29: "The evaluation of the model performance for SWE compared to GlobSnow indicates the same bias characteristics as described here for snow depth (not shown)." This is really confusing... please re-write**
Done. We deleted this sentence.

**P15 L5-9: "Across-model differences in the interannual variability of winter precipitation do not translate simply to corresponding differences in the interannual d$_{snow}$ variability (not shown)." This is another conclusion before the supporting statement. "For example, UVic calculates the (unrealistically) largest interannual dsnow variability in the boreal Europe permafrost region which is not reflected in the precipitation variability." This is the supporting statement.**
This is a subjective criticism of style rather than substance. There are countless examples in the scientific literature of similar styles as we adopt here, that is a general point followed by a specific example. We do think this paragraph is clear.

**P15 L13: "We have shown that the across-model spread in the representation of snow insulation effects". How did you show that? I think I lost something while reading Sections 3.1 and 3.2**
This done in Figs.1-5 as well as in Table2.

**P15 L15-16: "By considering the relationship plots and the conditional PDFs (Figs. 1, 2, 4, and 5)". Add figure #s here after "plots" and not all the figures in parentheses are related to PDFs.**
Done. *"By considering the relationship plots (Figs. 1, 4 and 5), and the conditional PDFs (Fig. 2) we were able to classify the models…"*

**P15 L17: "sort the models in terms of their snow insulation performance." Here "sort" means "classify"?**
Done. Yes, this is what was meant. For clarity, we replace "sort" by "classify".

**P15 L20-29: It is better to move this paragraph to the introduction, where the employed models are described.**
Sorry, we do not agree with you. Here we describe those specific model characteristics and processes which explain why these models show a better performance than the others. It makes no sense to list which models have performed better in the introduction before the analysis is done.

**P17 L1-2: "The results are further improved by updating the snow albedo and snow densification parameterization". References?**
Done. *"Decharme et al. (2016) still showed that the ISBA results are further improved by updating the snow albedo and snow densification parameterization."*

**P17 L20-32: Move this paragraph into the above sections.**
Sorry, we think this is a misunderstanding. This paragraph clearly belongs to the summary and conclusion. In this paragraph, we summarize the main findings.

**P18 L1: "The primary aim of this study was to …". I thought that "the aim" is stated on line 14, page 17. Please clarify.**
Done. We revised this sentence to *"This study uses the ensemble of models to*

*document model performance with respect to $T_{soil}$ versus $T_{air}$ relationships...*"

**P18 L9: "Those models which show limited skill in snow insulation representation …have some deficiencies". Maybe over simplifications?**
Done. We revised this to "*… have some deficiencies or oversimplifications...*"

**P18 L24-30: The manuscript has not discussed the modeled area of permafrost prior to this paragraph. The conclusions are to summarize primary findings. Please move this paragraph into the main part of this manuscript.**
Done. We agree with you and the other reviewer and we added an additional section "*5. Permafrost area*" in the results section in the paper with the additional new *Tab.4*, which presents the simulated permafrost area. "*Snow cover plays an important role in modulating the variations of soil thermodynamics, and hence near-surface permafrost extent (e.g., Park et al., 2015). Here we evaluate if there is a simple relationship between the simulated Northern hemisphere permafrost area and the sophistication and ability of the snow insulation component in the LSM to match observed snow packs. The simulated near-surface permafrost area varies greatly across the nine models in the hindcast simulation (1960-2009; Table 4). Some of the better performing snow insulation effect models (CLM4.5, JULES) simulate a near-surface permafrost area of 13.19 to 15.77 million $km^2$, which is comparable with the IPA map estimate (16.2 million $km^2$) (Brown et al., 1997; Slater and Lawrence, 2013). CoLM and ORCHIDEE, identified as reasonable models with respect to snow insulation, simulate much lower (7.62 million $km^2$) and higher (20.01 million $km^2$) areas, respectively. The main deficiency of CoLM is its too small soil depth (3.4 m) compared with CLM4.5 (45.1 m) despite having very similar snow modules (Table 1). However, ISBA, one of the two models that showed rather limited skill in representing snow insulation effects, also simulates the highest permafrost area (20.86 million $km^2$). This is inconsistent with previous studies (e.g., Vavrus, 2007; Koven et al., 2013) which concluded that the first-order control on modelled near-surface permafrost distribution is the representation of the air-to-surface soil temperature difference. Table 4 shows that the situation is more complex and that snow insulation simulation is not the dominant factor in a good permafrost extent simulation. When the land surface models having poor snow models are eliminated, the remaining models' simulated permafrost area show little or no relationship with the performance of the snow insulation component, because several other factors such as differences in the treatment of soil organic matter, soil hydrology, surface energy calculations, model soil depth, and vegetation also provide important controls on simulated permafrost distribution (e.g., Marchenko and Etzelmüller, 2013).*"
Accordingly, we shortened the permafrost part in the "6. Summary and conclusions" section. The according paragraph reads: "*Snow and its insulation effects are critical for accurately simulating soil temperature and permafrost in high latitudes. The simulated near-surface permafrost area varies greatly across the nine models (from 7.62 to 20.86 million $km^2$). However, it is hard to find a clear relationship between the performance of the snow insulation in the models and the simulated area of*

*permafrost, because several other factors e.g. related to soil depth and properties and vegetation cover also provide important controls on simulated permafrost distribution."*

**P26 L3: "Table 1. PCN snow model details." Please specify how the upper 20cm of the soil column are resolved in each model. How is the heat capacity is calculated, may add another column.**

According to our analysis, and as also noted by Referee #1, it is the snow layer in winter that is the key here, not the soil conditions (moisture, organic layer). Therefore, we think the current table contains enough information for the discussion in this paper, with its emphasis on snow above the soil. The general structure and some key parameterizations of snow processes can explain the main deficiencies in the simulated results. The interested reader may find such soil details in 3 papers we cite in the introduction on the model general characteristics.

**P28 L4: "The dots represent the medians of…". Please use different symbols. If this is printed black&white or someone is color blind, it might be hard to differentiate the results. P29 L3: "Conditional probability density functions…". Why are they conditional? "color". Same comments as for the above figure. Please think how to make this plots available to color-blind people. P30 L3: "Spatial maps of the correlation…"   Same comment as above P31 L4: "The dots represent the medians…" Same comment as above. Maybe use different symbols for dots. P32 L4: "The dots represent the medians…"**

We do indeed sympathize with colour-blind readers. Personally journal should do more to encourage such consideration in their publications, but this is clearly an issue at the journal publisher level, not at the author or editor level. Our figure presentation is consistent with what is common in TC and other journals.

"Conditional probability" is a completely standard term in statistics. The "conditional" means the $\Delta T$ PDFs are PDFs for specific different conditions, namely for different conditions of air temperatures and snow depths.

[revised manuscript text omitted]
=579) mean[St]= 26.4 cm, mean[GS]=23.4 cm stdev[St]= 9.0 cm, stdev[GS]= 6.5 cm | | | | | ΔT (n=268) mean[St]= 11.9 K stdev[St]= 2.3 K | | |
|---|---|---|---|---|---|---|---|---|
| | bias[St] | rmse[St] | bias[GS] | rmse[GS] | stdev | bias[St] | rmse[St] | stdev |
| CLM4.5 | 11.5 | 18.1 | 14.3 | 18.1 | 5.8 | 2.3 | 4.1 | 2.2 |
| CoLM | 15.6 | 21.4 | 17.8 | 22.1 | 9.8 | 2.7 | 3.7 | 2.4 |
| ISBA | 13.0 | 18.8 | 15.7 | 19.8 | 9.5 | -8.4 | 9.1 | 0.9 |
| JULES | -4.1 | 14.1 | -1.3 | 12.8 | 7.7 | -0.8 | 4.2 | 3.2 |
| LPJ-GUESS | -5.3 | 17.3 | -2.5 | 16.0 | 5.0 | -0.7 | 3.7 | 1.7 |
| MIROC-ESM | -0.4 | 17.9 | 1.9 | 14.0 | 6.3 | -4.9 | 6.7 | 2.0 |
| ORCHIDEE | -8.7 | 16.5 | -5.3 | 15.3 | 6.9 | -5.2 | 6.0 | 1.9 |
| UVic | -3.7 | 18.9 | -0.5 | 16.8 | 9.4 | -5.1 | 6.5 | 1.4 |
| UW-VIC | 12.5 | 19.8 | 15.0 | 20.0 | 10.4 | -1.3 | 4.8 | 2.1 |

**Table 4.** Permafrost area, defined as maximum seasonal active layer thickness < 3 m in 1960

(Mc Guire et al., 2016). The IPA map estimate is 16 million km$^2$ (Brown et al., 1997; Slater and Lawrence, 2013).

| Land Surface Model | Snow Insulation skill | Permafrost Area ($10^6$ km$^2$) |
|---|---|---|
| CLM4.5 | High | 15.77 |
| CoLM | High | 7.62 |
| ISBA | Low | 20.86 |
| JULES | High | 13.19 |
| LPJ-GUESS | Medium | 17.41 |
| MIROC-ESM | Medium | 13.02 |
| ORCHIDEE | Medium | 20.01 |
| UVic | Low | 16.47 |
| UW-VIC | Medium | 17.56 |

[Figure]

[Figure]

Figure 1. Variation of ΔT (K), the difference between soil temperature at 20 cm depth and air temperature) with snow depth (cm) for winter 1980-2000. The dots represent the medians of 5 cm snow depth bins and the upper and lower bars indicate the 25th and 75th percentiles, calculated from all Russian station grid points (n=268) and 21 individual winters. The numbers in each model panel indicate the RMSE between the observed and modeled relationship. Color represents different air temperature regimes.

[Figure]

**(a)**

[Figure]

[Figure]

**Figure 2.** Conditional probability density functions (PDFs) of ΔT (K), the difference between soil temperature at 20 cm depth and air temperature for (a) different snow depth classes and (b) air temperature regimes, for winter 1980-2000.for different snow depth and air temperature regimes (color) for winter 1980-2000.

[Figure]

**Figure 3.** Spatial maps of the correlation coefficients between snow depth and ΔT, the difference between soil temperature at 20 cm depth and air temperature for winter 1980-2000. Regions with greater than 95% significance are hashed.

[Figure]

[Figure]

**Figure 4.** Variation of soil temperature at 20 cm depth (℃) with air temperature (℃) for winter 1980-2000. The dots represent the medians of 5 ℃ air temperature bins and the upper and lower bars indicate the 25th and 75th percentiles, calculated from all Russian station grid points (n=268) and 21 individual winters. The numbers in each model panel indicate the RMSE between the observed and modeled relationship. Color represents different snow depth regimes.

[Figure]

[Figure]

**Figure 5.** Variation of soil temperature at 20 cm depth ( ℃; y axis) with snow depth (cm) for winter 1980-2000. The dots represent the medians of 5 cm snow depth bins and the upper and lower bars indicate the 25th and 75th percentiles, calculated from all Russian station grid points (n=268) and 21 individual winters. The numbers in each model panel indicate the RMSE between the observed and modeled relationship. Color represents different air temperature regimes.

[Figure]

**Figure 6.** Spatial maps of snow depth (cm) for winter 1980-2000.

---

## Referee Report (RR1)

[referee-annotated manuscript omitted]

---

## Author Response (AR2)

**Response to comments of editor:**
We thank the editor for providing additional specific comments/edits on our paper that helped further improving our manuscript. Please find **the editors comments in bold**, our point-by-point answers without formatting, and *changes to the initial manuscript in Italics*.

**Comment about replies**

**In your reply to P15 L13: "We have shown that the across-model spread in the representation of snow insulation effects". How did you show that? I think I lost something while reading Sections 3.1 and 3.2**
**You answered "*This done in Figs.1-5 as well as in Table2.*"**
**Did you address this comment also by clarifying the text? Please give further details.**
The across-model spread in the representation of the snow insulation effects is presented in both Sections 3.1 and 3.2. Therein the detailed discussion is with help of the Figures 1-5 as well as Table 2. This was a misunderstanding and we clarified this.

**Comments about revised paper**

**Page 1, line 15: the author's contribution should be added at the end of the paper, see also**
**http://www.the-cryosphere.net/for_authors/manuscript_preparation.html**
We add one paragraph for author contribution before the "Acknowledgement".
*"Author contributions: A.R. developed the idea that lead to this paper; W.W. and A.R. performed research; J.C.M. contributed with ideas; A.R., J.C.M., W.W., and D.J. wrote the paper; and all authors discussed the results and commented on the paper at all stages."*

**Page 2, line 5: sentence contains three times "bias", please rephrase**
Done. We revised this sentence with "Most of the model biases and across-model differences in simulating permafrost area are due to *inaccurate* atmospheric simulation e.g. of air temperature and precipitation, *deficient* simulation of snow and soil temperature and the coupling between atmosphere and land surface".

**Page 8, line 24: please correct sentence**
Done. We revised this sentence to "The modal value of the *station-based* $\Delta$ T PDF is 5 K for "shallow" snow and 14 K for "thick" snow…"

**Page 8, line 3: "Our analysis is focused on the common winter (DJF) condition, although snow can begin in November". Snow cover build up can start as early as September. Please state why you choose November.**
Yes, the first snow cover can build up in early autumn. The time of the first snow fall in the Arctic differ regionally and has a large inter-annual variability. But, the intention of our sentence is just to emphasize that a different winter definition does not change our results. Therefore, we investigated the common winter definition (DJF) and also another winter definition (NDJFMA) find in literature. However, to be more precise, we revised this sentence to "Our analysis is focused on the common winter (DJF) condition, although snow can begin in November *or even earlier*…"

In the sentence following the cited one we give the details about the investigated "snow parameters": "Most importantly, these studies showed that the consideration of wet snow metamorphism and snow compaction, improved snow thermal conductivity and multi-layer snow schemes can improve the simulation of snow dynamics and soil temperature. Parameterizations that take into account snow compaction (e.g. related to overburden pressure, thermal metamorphism and liquid water) work better than simpler schemes such as an exponential increase of density with time (Dutra et al., 2010). The influence of snow thermal conductivity on soil temperature…". We agree that "snow parameterization" describe this better. Accordingly, we have changed "snow parameter" to "snow *parameterization*".

**2."Other models (LPJ-GUESS, MIROC-ESM, ORCHIDEE) strongly underestimate the increase of the Tsoil vs. Tair regression slope for decreasing snow depth." Hard to understand.**
We change the sentence with *"Other models (LPJ-GUESS, MIROC-ESM, ORCHIDEE) do not reproduce the much greater regression slope between Tsoil vs. Tair for shallow snow than for thick snow as the observations show."*

**3."5 Permafrost area" This section is quite short compared to other sections. You may simply put to the above section and revise the section head as Discussion.**
Thanks for the suggestion, but the structure as it stands seems clearer to us. This section "5 Permafrost area" stands for its own and does clearly not fit into the other sections. Yes, it is smaller than the other sections, but we do not see any problem with this. It is shorter because this part is not the main part of the paper's analysis which focus on the air-soil temperature-snow relationships, but present the result that there is no clear relationship between the simulated permafrost area and 
[revised manuscript text omitted]